



# The YOPP site Model Intercomparison Project (YOPPsiteMIP) phase 1: project overview and Arctic winter forecast evaluation

Jonathan J. Day[1], Gunilla Svensson[2], Barbara Casati[3], Taneil Uttal[4], Siri-Jodha Khalsa[5], Eric Bazile[6], Elena Akish[4], Niramson Azouz[6], Lara Ferrighi[7], Helmut Frank[8], Michael Gallagher[4,9], Øystein Godøy[7], Leslie M. Hartten[4,9], Laura X. Huang[3], Jareth Holt[2], Massimo Di Stefano[7], Irene Suomi[10], Zen Mariani[3], Sara Morris[4], Ewan O'Connor[10], Roberta Pirazzini[10], Teresa Remes[7], Rostislav Fadeev[11], Amy Solomon[4,9], Johanna Tjernström[12], Mikhail Tolstykh[11,13],

[1]European Centre for Medium-Range Weather Forecasts, Reading, United Kingdom
[2]Department of Meteorology and Bolin Centre for Climate Change, Stockholm University, Sweden.
[3]Meteorological Research Division, Environment and Climate Change Canada, Canada.
[4]NOAA Physical Science Laboratory, Boulder, Colorado, USA.
[5]National Snow and Ice Data Center, University of Colorado,
[6]Meteo France, Toulouse, France
[7]Norwegian Meteorological Institute, Oslo, Norway.
[8]Deutscher Wetterdienst, Offenbach, Germany
[9]Cooperative Institute for Research in Environmental Science (CIRES), University of Colorado, Boulder, Colorado, USA
[10]Finnish Meteorological Institute, Helsinki, Finland.
[11]Marchuk Institute of Numerical Mathematics Russian Academy of Sciences, Russia
[12]Swedish Meteorological and Hydrological Institute, Linköping, Sweden
[13]Hydrometeorological Research Centre of Russia, Russia

*Correspondence to*: Jonathan J. Day (jonathan.day@ecmwf.int)

**Abstract.**

Although the quality of weather forecasts in the polar regions is improving, forecast skill there still lags the lower latitudes. So far there have been relatively few efforts to evaluate processes in Numerical Weather Prediction systems using in-situ and remote sensing datasets from meteorological observatories in the terrestrial Arctic and Antarctic, compared to the mid-latitudes. Progress has been limited both by the heterogeneous nature of observatory and forecast data but also by limited availability of the parameters needed to perform process-oriented evaluation in multi-model forecast archives. The YOPP site Model Inter-comparison Project (YOPPsiteMIP) is addressing this gap by producing Merged Observatory Data Files (MODFs) and Merged Model Data Files (MMDFs), bringing together observations and forecast data at polar meteorological observatories in a format designed to facilitate process-oriented evaluation.

An evaluation of forecast performance was performed at seven Arctic sites, focussing on the first YOPP Special Observing Period in the Northern Hemisphere (SOP1), February and March 2018. It demonstrated that although the characteristics of forecast skill vary between the different sites and systems, an underestimation in boundary layer temperature variance across models, which goes hand in hand with an inability to capture cold extremes, is a common issue at several sites. Diagnostic analysis using surface fluxes suggests that this is at least partly related to insufficient thermal representation of the land-surface in the models, which all use a single layer snow model.

## 1 Introduction

Recent decades have seen a marked increase in human activity in the polar regions leading to an increasing societal demand for weather and environmental forecasts (Emmerson and Lahn, 2012; Goessling et al., 2016). Despite this growing need, the skill of weather forecasts in the polar regions lags that of the mid-latitudes (Jung et al., 2016; Bauer et al., 2016). This is partly the result of the relatively lower density of conventional observations in high compared to mid-latitudes (Lawrence et al.,



2019), but is also related to the occurrence of meteorological situations and phenomena which are historically difficult to model such as stable boundary layers (e.g. Atlaskin and Vihma, 2012; Sandu et al., 2013; Holtslag et al., 2013), mixed-phase clouds (e.g. Pithan et al., 2014, 2016, Solomon et al., 2023), and the importance of coupling between the atmosphere and snow and ice surfaces (e.g. Day et al., 2020; Batrak and Muller, 2019; Svensson and Karlsson, 2011).

The ability of climate models to represent atmospheric processes in polar regions has recently been assessed highlighting deficiencies in near-surface and boundary layer properties (Pithan et al., 2014; Svensson and Karlsson, 2011; Karlsson and Svensson, 2013). Since many climate models are based on global weather forecasting systems, understanding the causes of forecast error after 1-2 days may help develop understanding of the sources of error in climate models (Rodwell and Palmer, 2007). Nevertheless, until recently there has been little focus on evaluating Numerical Weather Prediction (NWP) models

using in-situ data from the terrestrial Arctic and Antarctic (Jung and Matsueda, 2014; Jung et al., 2016).

Recent studies, conducted as part of the World Weather Research Programme's Polar Prediction Project (PPP, Jung et al, 2016) have started to address this gap, assessing the skill of both the large scale circulation (Bauer et al., 2016) and surface weather properties (Køltzow et al., 2019). The Year of Polar Prediction (YOPP) site Model Intercomparison Project

(YOPPsiteMIP) was designed to build on these earlier studies by utilising process level data from polar observatories to diagnose the causes of forecast error from a process perspective and ultimately inform model development. Although process-oriented evaluation studies focussing on polar processes are not new, those that have been done have tended to focus on one or two sites or a specific field campaign (see Day et al., 2020; Batrak and Müller, 2019; Miller et al., 2018; Tjernström et al., 2021 for some recent examples). A key aim of YOPPsiteMIP is to provide a pan-Polar perspective on forecast evaluation and

process representation.

YOPPsiteMIP participants were asked to provide data in so-called Merged Data Files (MDFs) which includes both Merged Observatory Data Files (MODFs), for observatory data, and Merged Model Data Files (MMDFs), for model data. These data standards, which were developed specifically for YOPPsiteMIP, are described by Uttal et al. (2023). Using this common file

format, with consistent naming and metadata, facilitates equitable and efficient comparisons between models and observations. This standardisation of the data from different observatories also aids interoperability in the sense that the same evaluation code can be applied at different sites. These MDF filetypes were developed as part of PPP, following the FAIR (Findable, Accessible, Interoperable, Reusable) data principles (Wilkinson, 2016). Details of the MDF concept and specifics of the data processing chain and related Python toolkit for producing MDFs are described in Uttal et al. (2023) and Gallagher et al., (in

prep).

The observatories selected for YOPPsiteMIP represent a geographically diverse set of locations. At these sites a wide range of instruments measuring properties of the air, snow and soil are employed, extending far beyond the traditional synoptic surface and upper-air observation network, which are collected for use in the production and evaluation of NWP systems (Uttal et al.,

2015). Taken together, the observations collected at these observatories offer opportunities to develop a deeper understanding of the physical processes governing the weather in the polar regions, their representation in forecast models, and how this varies from site to site. The processes and phenomena targeted in YOPPsiteMIP include boundary-layer turbulence, surface exchange (including over snow and ice) and mixed-phase clouds.

A benefit of organizing coordinated evaluation involving several NWP systems and multiple sites is that it helps clarify if the issues revealed by the analysis are model or location specific. The modelling community has organized model inter-comparisons to target various atmospheric processes relevant for Arctic conditions (e.g. Cuxart et al., 2006; Pithan et al., 2016;

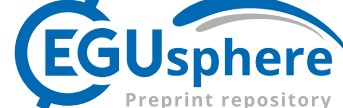

Tjernström et al 2005, Sedlar et al. 2020, Solomon et al., 2023) each using its own protocol for data sharing. However, the newly developed standardisation of the observational and forecast model data developed for YOPPsiteMIP is planned to be

used for future MIIPs (model intercomparison and improvement projects). Converging on a standard like this will aid interoperability, making it easier for model developers to expand their evaluation to new sites or observational campaigns, but also to other models or forecasting systems.

MMDFs were requested for the locations listed in Table 1 and shown in Figure 1 during the YOPP Special Observing Periods,

during which the observations taken at many polar observatories (e.g. the frequency of radiosondes) was enhanced (see Lawrence et al., 2019; Bromwich et al., 2020). For the Northern Hemisphere the periods Feb–Mar 2018 and Jul–Sep 2018 were selected and named NH-SOP1 and SOP2 respectively. For the Southern Hemisphere or SH-SOP the period Nov–Feb 2018/19 was chosen.

| Observatory name Filename | Latitude Longitude | Elevation |
|---|---|---|
| **Arctic land sites** | | |
| Utqiaġvik (Formerly known as Barrow, Alaska) *Utqiaġvik* | 71.32°N, 156.62°W | 8-20 m |
| Oliktok Point (Alaska) *oliktok* | 70.50°N 149.89°W | 2-6 m |
| Whitehorse (Canada) *whitehorse* | 60.71°N, 135.07°W | 682 m |
| Eureka (Canada) *eureka* | 80.08°N 86.42°W | 0-610 m |
| Iqaluit (Canada) *iqaluit* | 63.74°N, 68.51°W | 5-11 m |
| Alert (Canada) *alert* | 82.49°N, 62.51°W | 8-210 m |
| Summit (Greenland) *summit* | 72.58°N, 38.48°W | 3210-3250 m |
| Ny-Ålesund (Svalbard) (Zeppelin station) *nyalesund* | 78.92°N, 11.53°E (78.9°N, 11.88°E) | 0-30 m (473 m) |
| Sodankylä (Finland) *Sodankylä* | 67.37°N, 26.63°E | 198 m |
| Pallas (Finland) *pallas* | 67.97°N, 24.12°E | 305 m |





| Tiksi (Russia) *tiksi* | 71.60°N, 128.89°E | 1-30 m |
|---|---|---|
| Cherskii (Russia) *cherskii* | 68.73°N, 161.38°E (68.51°N, 161.53°E) | 8 m (16 m) |
| Ice Base Cape Baranova (Russia) *baranova* | 79.3°N, 101.7°E | 24 m |

**Arctic Ocean sites**

| SHEBA location *sheba* | 165°W, 76°N | Sea level |
|---|---|---|
| Arctic Ocean 1 (Gakkel Ridge) *ao1* | 10°E, 85°N | Sea level |
| Arctic Ocean 2 (North Pole) *ao2* | 0°E, 90°N | Sea level |
| Arctic Ocean 3 (Canada Basin) *ao3* | 135°W, 81°N | Sea level |

**Antarctic land sites**

| Alexander Tall Tower *alexander* | 79.01°S, 170.72°E | 55 m |
|---|---|---|
| Casey *casey* | 66.28°S, 110.53°E | 30 m |
| Davis *davis* | 68.58°S, 77.97°E | |
| Dome C *domec* | 75.08°S, 123.34°E | 3233 m |
| Dumont d'Urville *dumont* | 66.66°S, 140.01°E | 0-50 m |
| Halley IV *halley* | 75.58°S, 26.66° W | 130 m |
| King Sejong (King George Island) *kingsejong* | 62.22°S, 58.79° W | 10 m |

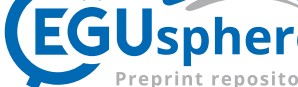

| Georg von Neumayer | 70.65°S, 8.25°W | 42 m |
| *neumayer* | | |
| Mawson | 67.60°S, 62.87°E | 15 m |
| *mawson* | | |
| Syowa (Showa) | 69.00°S, 39.59°E | 18-29 m |
| *syowa* | | |
| Jang Bogo (Terra Nova Bay) | 74.62°S, 164.23°E | 36 m |
| *jangbogo* | | |
| Amundsen-Scott South Pole | 90°S, 0°E | 2835 m |
| *southpole* | | |
| Byrd | 80.01°S, 119.44°W | 1539 m |
| *byrd* | | |
| Rothera | 67.57°S, 68.13° W | 4 m |
| *rothera* | | |
| Vostok | 78.46°S, 106.84°E | 3489 m |
| *vostok* | | |
| McMurdo | 77.85°S, 166.67°E | 10 m |
| (Scott base) | (77.85°S, 166.76°E) | (10 m) |
| *mcmurdo* | | |
| Troll | 72.01°S, 2.54°E | 1275 m |
| *troll* | | |

Table 1: List of YOPPsiteMIP observatory locations: name, *name as used in filenames,* latitude, longitude and elevation. Where an elevation range is stated, this is because the instruments at a given observatory extend over a range of values due to variations in local topography.




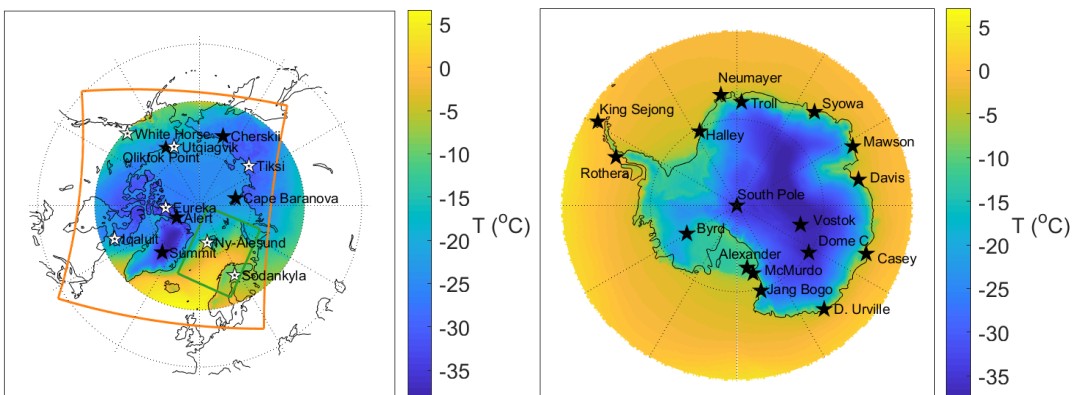

**Figure 1: Maps of the ERA5 2m-temperature climatology (1990-2019) for February-March (time of NH-SOP1) for Arctic (left) and for November-February (SH-SOP) for Antarctic (right). The observatories used in YOPPsiteMIP are marked with stars. White stars indicate the sites where MODFs are currently available, which are the subject of this study; black stars indicate the sites whose MODFs are not yet complete. The orange and green boxes depict the extent of the ECCC-CAPS and AROME-Arctic domains respectively.**

The purpose of this paper is two-fold: firstly, to document the first version of the YOPPsiteMIP dataset along with a basic description of the forecasting systems and their respective MMDFs that are archived at the YOPP Data Portal, hosted by the Norwegian Meteorological Institute (MET Norway). Secondly, the paper presents a multi-site evaluation of seven forecasting systems during NH-SOP1, at seven Arctic observatories that have produced MODFs. The locations are indicated by the white stars in Figure 1a and the MODFs and details of the sites are described in Morris et al., (in prep).

## 2    Description of simulations, model formulation and output protocol

To date, six NWP centres have submitted forecasts from seven forecasting systems for SOP1 & SOP2, with two systems submitted for the SH-SOP (see Table 2).  Four of the systems are global:

- The Integrated Forecasting System from the European Centre for Medium-Range Weather Forecasts (ECMWF-IFS; Day et al., 2023),
- The Action de Recherche Petite Echelle Grande Echelle from Meteo France (ARPEGE-MF ; Bazile and Azouz, 2023a),
- The Semi-Lagrangian, based on the absolute vorticity equation from the Hydrometeorological Research Centre of Russia (SLAV-RHMC, Tolstykh, 2023) and,
- The Icosahedral Nonhydrostatic Model from Deutscher Wetterdienst (DWD-ICON; Frank, 2023).

Three are regional:

- The Canadian Arctic Prediction System from Environment and Climate Change Canada (ECCC-CAPS; Casati, 2023)
- and two versions of Applications of Research to Operations at Mesoscale (AROME) from Meteo France (AROME-MF; Bazile and Azouz, 2023b) and from MET Norway (AROME-Arctic; Remes, 2023).

The domain boundaries of the regional forecasting systems can be seen in Figure 1 (note that only two of the observatories are within the AROME domain). The forecasts analysed here were initialised at 00 UTC for each day of the SOPs (although 12UTC forecasts are also available on the archive for many of the systems). The forecast leadtime varies between the different systems but all forecasts are at least two days long (see Table 2 and Figs 2 & 3).

The files for some of the systems (CAPS, SLAV, ARPEGE, AROME-MF) are provided with multiple grid-points, centred on the observatory location. For others only a single grid-point was provided. Multiple grid-points centred around the observatory



location were requested because many of the observatories are located on coasts and there are issues with grid-points being over ocean, over land or blended. In this study when there are multiple grid points we choose the closest 100% land point to the supersite location. The grid resolutions range from 2.5 km to ~30 km and the model timestep varies from 1.5 to 7.5 min (see Table 2).

The models have quite a diverse mixture of formulations for atmospheric dynamics, land surface, sub-grid scale parameterisations and initialisation/data assimilation procedures. More details about the simulations with specific models are provided below and a summary of the key model components/parameterisations used in each model is included in Table 3.

## 2.1 IFS-ECMWF

MMDFs for the operational forecasts with the IFS high resolution deterministic forecasts are available for the period starting Jan 2018. The initial forecasts are produced with IFS cycle 43r3 which was an atmosphere only model with persisted sea ice and anomaly SSTs. From 5 June 2018 (i.e. before SOP2) the forecasts were produced with cycle 45r1 which included dynamic sea ice and ocean fields (see Day et al., 2022 for more information). Although the model version changes the horizontal (~9km) and vertical resolution (L137) are the same in all SOPs. The data archived in the MMDFs is provided at the model timestep

(7.5 min) for a single model grid point closest to the observatory. In addition to the grid point data a number of parameters (including albedo, surface temperature and surface energy fluxes) are provided on the land-surface model tiles to enable detailed evaluation of processes even at heterogeneous sites. A complete description for the two versions of the IFS can be found here: https://www.ecmwf.int/en/publications/ifs-documentation.

## 2.2 ARPEGE-MF

The version of ARPEGE submitted to YOPPsiteMIP was a pre-operational version based on the cy43t2_op1 operational system but coupled with the 1D sea-ice model GELATO (Bazile et al. 2020). The resolution of the model used for these simulations is the same as is used operationally at Meteo France which is variable (using a stretching factor of 2.2) with the pole (highest resolution of 7.5 km) over France for SOP1 and SOP2 and over Antarctica in SOP-SH and 105 vertical levels.

The horizontal resolution is about 8-9 km over the North-Pole and timeseries have been provided for the three SOPs in the MMDF format for the 21 YOPP observatories with an hourly output for both state variables (instantaneous) and fluxes (accumulated).

## 2.3 SLAV-HMRC

MMDFs were produced by the SLAV model (Tolstykh et al., 2018) for both SOP1 and SOP2 containing 7-day forecasts starting at 00 UTC. The output is available for 4 horizontal grid points surrounding selected observatories, every 15 minutes (i.e. every fourth timestep). Depending on variable, the output is instantaneous or a 15-min averaged value. Data for 13 of the Arctic observatorys in Table 1 are provided. Selection of observatories is based on model resolution in latitude which is relatively low, ~16 km in Northern polar areas; also, the ao2 point is not included because the model grid does not contain the

poles.

## 2.4 ICON-DWD

MMDFs from DWD's ICON (Zängl et al., 2015) are available from February 2018 to June 2020 containing 7.5-day forecasts starting at 00 and 12 UTC for Sodankylä, Ny-Ålesund, and Utqiaġvik (Barrow). The mesh width is 13 km. Different model

versions are used during this period. In February icon-nwp-2.1.02 was used followed by icon-2.3.0-nwp0 during 2018-02-14 to 2028-06-06, and from 2018-09-19 to 2018-12-05 icon-2.3.0-nwp2 was in operation. Since 2018-02-14, a new orographic data set came in operations, however, for the 3 data points provided the changes were less than 1 m in height. The sea ice



analysis used in ICON, was based on the Real-Time Global SST High Resolution Analysis of NCEP until 2018-07-16. Since then it is based on the Operational Sea Surface Temperature and Sea Ice Analysis (OSTIA; Donlon et al., 2012). To represent

variations of subgrid scale surface characteristics ICON uses a tile approach. Since 2018-07-16 the tile values of surface fluxes, and other tile dependent variables are included in the MMDFs in addition to the grid average values. Hourly output is available based on a timestep of 120s.

### 2.5 CAPS-ECCC

MMDFs for ECCC-CAPS are available for the whole period from February 2018 to December 2018. Prior to the 28th of June 2018 CAPS was uncoupled and run with the GEM version 4.9.2. After the 29th of June 2018 CAPS was coupled with the Regional Ice and Ocean Prediction system (RIOPS) and run with the GEM version 4.9.4. Atmospheric Lateral Boundary Conditions (LBCs) and initial conditions (ICs) are from ECCC Global Deterministic Prediction System (GDPS). Initial surface fields are from the Canadian Land Data Assimilation System (CaLDAS). The CAPS timeseries are produced for a beam of 7

x 7 grid-points centred on each of the twelve land-based Arctic observatories listed in Table 1. Timeseries up to 48 hours leadtime are made available for the daily runs initialized at 00 UTC. The data is archived with a time frequency of 7.5 min, equivalent to five timesteps of 90 s each.

### 2.6 AROME-ARCTIC

MET Norway utilises the HARMONIE-AROME (HIRLAM–ALADIN Research on Mesoscale Operational NWP in Euromed–Application of Research to Operations at Mesoscale) model configuration (Bengtsson et al., 2017) for operational weather forecasting for the European Arctic with the name AROME-Arctic (Muller et al., 2017). AROME-Arctic MMDFs are based on the operational forecasts (cy40h.1) and are available for the SOP1 and SOP2 at Sodankylä and Ny-Ålesund. LBCs are derived from the ECMWF IFS-HRES described in Section 2.1. Assimilation of conventional and satellite observation with

3DVAR in the upper atmosphere, optimal interpolation of snow depth, screen level temperature and relative humidity in the surface model. Temperature tolerance in the surface assimilation scheme was increased on 15 March 2018 to better assimilate observed low temperatures. The data archived in the MMDFs are provided hourly for the single model grid-point closest to the site. Model data for the full domain in its original format are also available via thredds.met.no.

### 2.7 AROME-MF

The AROME -MF system from Météo-France and AROME-ARCTIC from MET Norway are both configurations of the same model system but use different parameterizations of turbulence, shallow convection, cloud microphysics and sea ice. The system used for the YOPPsiteMIP differs from the operational AROME-France configuration (Seity et al., 2011) and the version evaluated for SOP1 in Køltzow et al., (2019) in that it is coupled with the GELATO 1D sea ice model. However, the

domain (see Figure 1a), horizontal and vertical grid are exactly the same as the AROME-ARCTIC operational system (see Section 2.6). The ICs and LBCs are interpolated from the global model ARPEGE-MF simulation descried above (Section 2.2). The MMDF files have been produced for Ny-Ålesund, Sodankylä and Pallas with hourly output.

### 2.8 Output format

For each forecast initial time and each forecasting system a single netCDF file containing all variables was archived following the MMDF format, which use the same nomenclature, metadata, and structure as the MODFs. In order to be able to assess process representation, the YOPPsiteMIP protocol requested that atmospheric fields were provided on native model vertical levels and all fields should be provided with high frequency (every 5 or 15 minutes), ideally at the frequency of the model timestep if practical.




The actual variables archived, frequency and number of grid-points, vary from model to model. For example, ECCC provided a comprehensive set of parameters for the CAPS model focusing on precipitation and clouds microphysics to allow studies on the representation of different types of hydrometeors by the P3 scheme (Morrison and Milbrandt, 2015; Morrison et al., 2015; Milbrandt and Morrison, 2016). A full list of requested variables, along with a schema for producing the MDFs are described

in a document known as the H-K Table (Hartten and Khalsa, 2022). The table is available in both human and machine-readable form (PDF and JSON, respectively). The H-K Table relies on standards and conventions commonly used in the earth sciences, including netCDF encoding with CF naming and formatting conventions and is an evolving document that is expected to evolve to fulfil the requirements of future MMDFs and MODFs (Gallagher et al., in prep.). The prescribed metadata make data provenance clear and encourage proper attribution of data origin (see further information in Uttal et al., 2023).


Although we only focus on model performance during SOP1, a full set of MMDFs and MODFs was produced for both SOPs. The MODFs for Iqaluit (Huang et al., 2023a), Whitehorse (Huang et al., 2023b), Utqiaġvik (formerly known as Barrow: Akish and Morris, 2023a), Eureka (Akish and Morris, 2023b), Tiksi (Akish and Morris, 2023c), Ny-Ålesund (Holt, 2023) and Sodankylä (O'Conner 2023) are described in detail in Morris et al., (2023) along with descriptions of the site geography.

MMDFs have also been produced for the SH-SOP with the ECMWF-IFS and ARPEGE models (See Table 2), but no MODFs for the Antarctic observatorys have been produced yet.

| Centre | Model-name | Global/Regional and horizontal/vertical resolution | Timestep/output frequency/forecast length | Version | Key Reference(s) | SOPs in YOPP portal |
|---|---|---|---|---|---|---|
| ECMWF | IFS | Global: 9km/L137 | 7.5min/7.5min/3d | Cy43r3 for SOP1, Cy45r1 for SOP2 & SOP-SH | Buizza et al., (2017) | SOP1, SOP2 & SOP-SH |
| Meteo-France | ARPEGE-GELATO | Global: 7.5-25km/L105 | 240s/60min/4d | cy43t2_op2 | Pailleux et al. (2014) | SOP1, SOP2 & SOP-SH |
| Meteo-France | AROME-Arctic | Regional: 2.5km/L65 | 50s/60min/2d | cy43t2_op2 | Seity et al., (2011) | SOP1 & SOP2 |
| ECCC | CAPS | Regional: 3km/L62 | 1.5min/7.5min/2d | vn1.0.0 for SOP1 & vn1.1.0 for SOP2 | Milbrandt et al., (2016)<br><br>Casati, et al., (2023) | SOP1 & SOP2 |





| DWD | ICON | Global: ~13km/L90 | 2min/60min/7.5 d | icon-nwp-2.1.02, icon-2.20-nwp0, icon-2.30-nwp0, icon-2.30.nwp2 | Zängl et al., (2015)<br><br>Prill et al., (2020) | SOP1 & SOP2 |
|---|---|---|---|---|---|---|
| HMCR | SLAV | Global: ~20km/L51 | 3.75min/15min/ 3d | SLAV20 (2018) | Tolstykh et al., (2018)<br><br>Tolstykh et al., (2017) | SOP1 & SOP2 |
| MET Norway | AROME-Arctic | Regional: 2.5km/L65 | 50s/60/2d | HARMONIE-AROME cy40h | Müller et al. (2017)<br><br>Bengtsson et al., (2017) | SOP1 & SOP2 |

**Table 2. Summary of forecasting systems**



| Model-name | Land-surface model | Surface layer/Fluxes | Turbulent diffusion | Orographic drag | Convection | Cloud microphysics | Radiation | Dynamical core |
|---|---|---|---|---|---|---|---|---|
| IFS | HTESSEL: Balsamo et al., (2009) | K-diffusion with stability functions of Dyer (1974) and Högström (1988) and Holtslag and De Bruin (1988) in unstable conditions and for stable conditions | EDMF Köhler et al., (2011) in unstable conditions and K-diffusion (Louis, 1979; Sandu et al., 2013) in stable conditions | Following Lott and Miller (1997) and Baines and Palmer (1990) | mass-flux for deep, shallow and mid-level convection: Tiedtke (1993) and Bechtold et al. (2008) | double moment scheme with four categories of hydrometeor Forbes and Ahlgrimm (2014) | EcRad (Hogan and Bozzo, 2018) is based on the Rapid Radiation Transfer Model (RRTM, Mlawer et al., 1997; Iacono et al., 2008) | Spectral/FE/H |
| ARPEGE | SURFEX: Masson et al., (2013) | K-diffusion with modified version of Louis (1979) | TKE: Cuxart et al., (2000) with a modified mixing length (Bazile et 2011) | Scheme described in Catry et al., (2008) following Lott and Miller (1997) for gravity wave drag, and envelope approach (after Wallace et al., 1983) | Mass flux for deep convection following Bougeault (1985) and mass flux for shallow convection following Bechtold orography et al., (2001) | Single moment with five categories of hydrometeor (Siety et al., 2012) | RRTM | Spectral/FE/H |
| AROME-MF | SURFEX: Masson et al., (2013) | K-diffusion with stability function of Louis (1979) | TKE: Cuxart et al., (2000) | N/A | Deep convection is explicitly represented and shallow uses the Pergaud et al. (2009) EDMF scheme. | Single moment with six categories of hydrometeor (ICE3; Pinty and Jabouille 1998) | RRTM | Spectral/FD/NH |
| CAPS | ISBA: Noilhan and Planton (1989) and Bélair et al. (2003) | K-diffusion with stability functions of Delage and Girard (1992) in unstable conditions and Delage (1997) in stable conditions. | TKE with statistical representation of subgrid-scale cloudiness (MoisTKE: Bélair et al. (2005)) | Lott and Miller (1997) | Deep convection from the Kain and Fritsch (1990) mass flux scheme and shallow convection from a Kuo-transient scheme (Bélair et al., 2005) | Double moment with Predicted Particle Properties (P3; Morrison and Milbrandt, 2015; Morrison et al, 2015; Milbrandt and Morrison, 2016) | Correlated-k distribution radiative transfer scheme (Li and Barker, 2005) | Gridpoint/FE (horizontal)&FD(vertical)/N H (Coté et al, 1998a,b; Girard et al, 2014) |





| | | | | | | | | |
|---|---|---|---|---|---|---|---|---|
| ICON | TERRA: Heise et al., (2006) | transfer-resistances approach: Baldauf et al., (2011) | TKE Baldauf et al., (2011) and Raschendorfer (2001) | Lott and Miller (1997) | mass-flux for deep, shallow and mid-level convection: Tiedtke (1993) and Bechtold et al. (2008) | Single moment scheme with four hydrometeors (Seifert, 2008) | RRTM | Grid-point/FV/NH |
| SLAV | ISBA 2L: Noilhan and Planton (1989) with modifications | Stability functions based on Cheng et al. (2002) with modifications leading to the absence of critical gradient Richardson number in the system. | TOUCANS (TKE+TTE) (Bastak-Duran et al 2014) | Scheme described in Catry et al., (2008) following Lott and Miller (1997) for gravity wave drag, and an envelope orography approach (after Wallace et al., 1983) | Mass flux following Bougeault (1982) but with modifications according to Gerard and Geleyn (2005) | Single moment scheme with four hydrometeors (Gerard et al., 2009) | Shortwave radiative transfer uses the CLIRAD model (Tarasov and Fomin, 2007) and RRTM for longwave | Grid-point/FD/H Tolstykh et al., (2017) |
| AROME-Arctic | SURFEX: Masson et al. (2013) | Based on Louis (1979) | HARATU: TKE together with a diagnostic length scale (Lenderink and Holtslag 2004; van Meijgaard et al. 2012) | N/A | Deep convection is explicitly represented and Shallow is represented by EDMF (Soares et al. 2004; Siebesma et al. 2007, Bentsson et al. 2017) | Single moment with five categories of hydrometeor based on Pinty and Jabouille (1998) with modifications (Müller et al 2017) | RRTM (EcRad) With modified cloud optical properties compared to AROME-MF (Bengtson et al. 2017) | Spectral/FD/NH |

**Table 3.** Details of physical processes and parameterizations of the forecasting systems (see Appendix A for list of acronyms).



### 3 Evaluation of basic surface meteorology and vertical profiles

#### 3.1 Evaluation/Scores

As mentioned in the introduction, the combination of MODFs and MMDFs allow detailed process-oriented diagnostics to be performed for the models. However, it is first important to assess what the errors are for standard variables such as 10m wind speed and 2m temperature. This first step is important because if they are stationary with leadtime one can simply consider a 24hr time range in the forecasts such as T+25 until T+48 (the second day of the forecast), simplifying the analysis.

The 2m temperature errors have quite different properties at each site and for each model (Fig 2). The models are typically too warm at Utqiaġvik and Tiksi and too cold at Ny-Ålesund and Whitehorse, with the sign of the bias varying between the models at Iqaluit and Eureka. At both Sodankylä and Whitehorse, which are situated at lower latitudes than the other sites, there is a distinct diurnal cycle in the bias and standard deviation that is not there at higher latitude sites. At both sites the night-time temperature bias is typically more positive than the daytime bias, indicating an underestimate of the diurnal temperature range.

In the case of the CAPS and the IFS, the bias in the diurnal cycle at these observatories are representative of those seen over wider region (e.g. Casati et al., 2023 and Haiden et al., 2018).

In terms of wind speed, the forecasts all have a positive wind speed bias at Utqiaġvik and a negative bias at Iqaluit and Whitehorse (Fig 3). At Tiksi, Eureka, Sodankylä and Ny-Ålesund, the sign of the bias varies between the models. Interestingly,

the largest inter-model spread and biases in wind speed is observed at the sites with the most complex orography (i.e. Iqaluit, Ny-Ålesund, Eureka and Tiksi: see Fig 2 of Morris et al., 2023), likely due to the difficulties in representing the mesoscale flow patterns typically generated in such locations. Interestingly, there does not seem to be an obvious benefit from the increased resolution, with the AROME configurations and CAPS model actually having worse biases than the lower resolution global models at Ny-Ålesund.


Although there is some sub-daily variability with a diurnal frequency in the bias, more pronounced in wind speed bias (Figs. 2 and 3), the size of the biases does not grow dramatically with time. Thus, we consider a 24hr time range between the T+25 and T+48 forecast steps (i.e. the second day of the forecast) to be representative of the general error, simplifying the analysis.


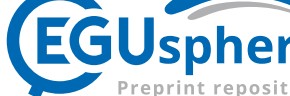

**Figure 2: Mean bias (solid lines) and standard deviation (dashed lines) of the 2m temperature error (in °C) at each observatory (see Figure 1a) for forecasts initialised at 00z during SOP1, described in Table 2.**






**Figure 3: Mean bias (solid lines) and standard deviation (dashed lines) of the 10m wind speed error (in m s⁻¹) at each observatory for forecasts initialised at 00z during SOP1.**





### 3.2 Vertical profiles

To gain further insights we investigate the vertical structure of the errors by comparing the model output to observations from radiosonde and tower. The median temperature and specific humidity within the boundary layer is overestimated at Tiksi, Eureka, Utqiaġvik and Iqaluit (see Fig 4) and the models underestimate the strength of temperature and humidity inversions as a result. The picture is more mixed at Ny-Alesund and Sodankylä where all models are too cold, and two out of the three models are too dry at Whitehorse.


   The biases in the upper air temperatures, 2m air temperature, and the surface skin temperature tend to go hand-in-hand with each other, i.e. model with warmest/coldest surface temperature tends to have the warmest/coldest 2m and upper air temperatures. As a result, the mean 2m temperature errors seen in Fig 2 give a sense of the sign of the error in the lowest 100m, or so, of the atmosphere. This coupling between the lowest model level, the surface skin temperature and the 2m-temperature

is to be expected, since the 2m-temperature is a diagnostic calculated as a function of the lowest atmospheric model layer and the surface skin temperature.

   Air temperature variability in the lower boundary layer is generally underestimated by the models, except at Iqaluit (Fig 5). This generally translates to an underestimation of the 2m temperature variability at these sites. Interestingly, at Ny-Alesund

some models severely overestimate the 2m temperature variance despite underestimating the variance aloft. For specific humidity the observed inter-quartile-range tends to sit within the range of the models, however it is over-estimated at Eureka and underestimated at Tiksi and Whitehorse in the lower boundary layer.

   The median of the modelled wind speed is too high in the boundary layer at Sodankylä, Utqiaġvik and Tiksi, but more mixed

at other sites (Fig 4 & 5). The variance of the wind speed is within the model range, with the exception of Iqualuit, where it is underestimated. The overestimation of the wind speed at these sites is likely a contributing factor in the underestimation of the temperature and humidity inversions, since a positive bias in the wind speed will drive excessive turbulent mixing of heat and moisture inhibiting the decoupling of near-surface and upper air temperatures that occurs during periods of radiative surface cooling and low wind (Van de Weil et al., 2017). Other factors which could play a role are the radiative forcing at the surface

or the response of the surface to radiative forcing. Both aspects will be addressed in the following subsection.





**Figure 4: Median temperature (left), specific humidity (middle) and wind speed (right) from the radiosonde (black solid line), the tower (black dashed line), and the numerical models (during the second day of the forecast: colour lines). The mean surface skin temperature is indicated by a dot, 2m temperature (left), 2m specific humidity (middle) and**
**10m wind speed (right) are shown with a square. Note that wind speed and humidity profiles from the tower are not available in the Tiksi and Ny-Ålesund MODFs respectively. The numbers in the left hand panels correspond to the verification sample size, which was dictated by the availability of radiosonde profiles.**




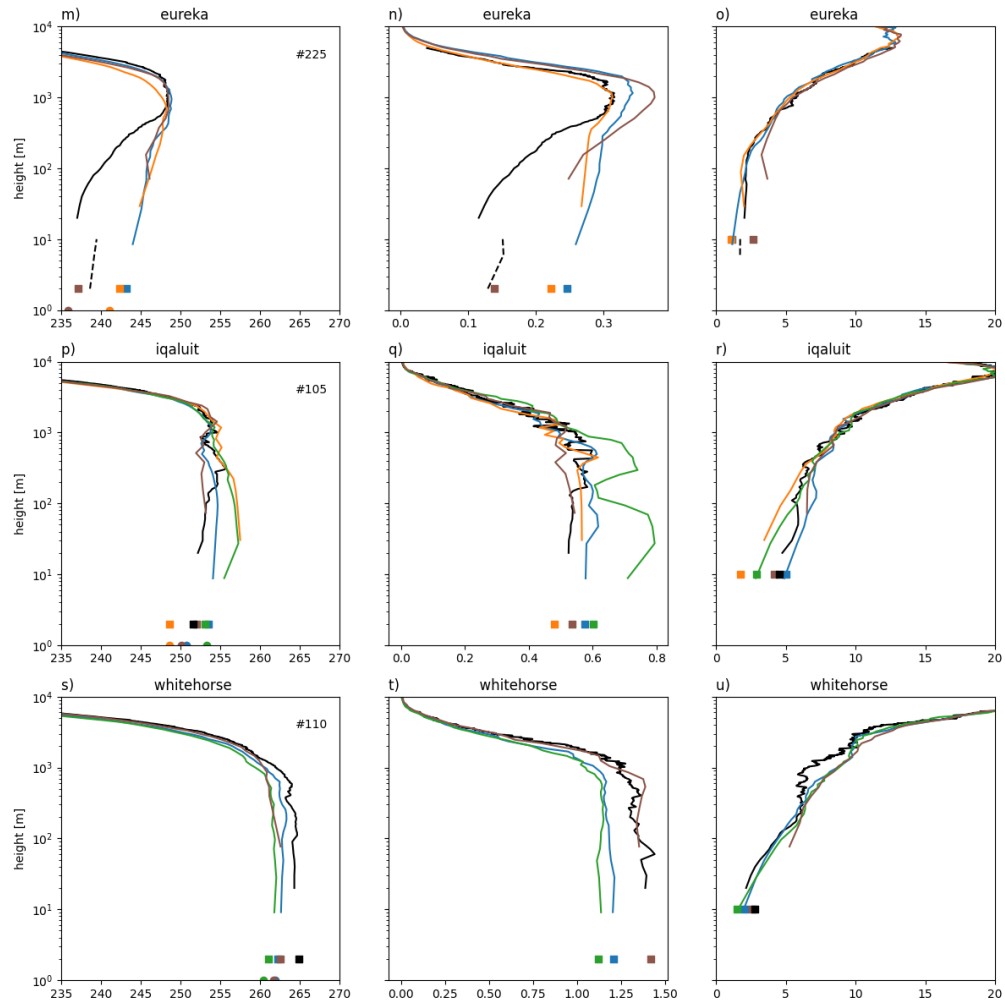

**Fig 4 continued.**






**Figure 5: As Figure 4 but showing the Inter Quartile Range.**



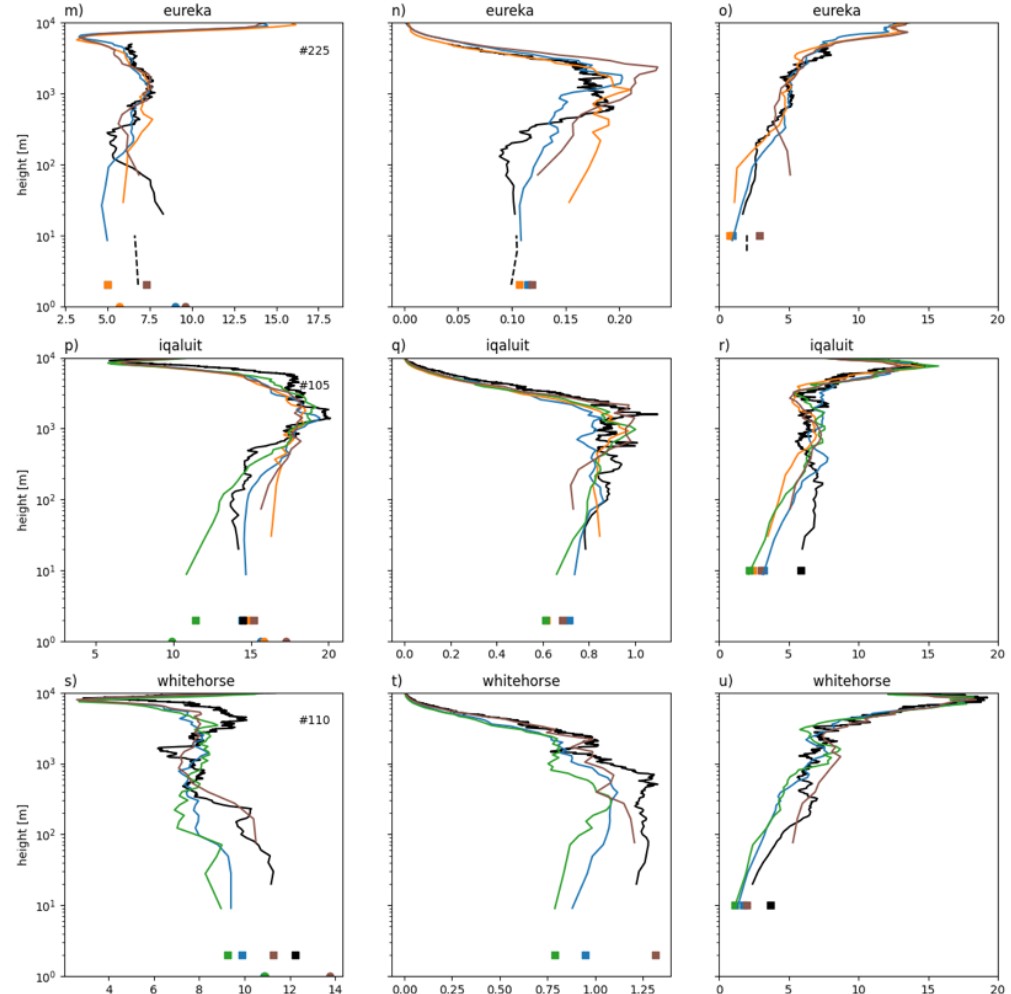


**Fig 5 continued.**

**3.3  Links between errors in boundary-layer temperature variance and surface radiation.**

In this section we investigate the role of radiative forcing in the underestimation of near-surface and boundary-layer

temperature variability at Sodankylä, Utqiaġvik and Tiksi where the models underestimate the temperature variability. At these sites all upwelling and downwelling radiation components are available in the SOP1 MODFs allowing us to investigate whether the suppressed temperature variability is related to supressed variability in the radiative forcing at the surface, a lack of sensitivity of the near-surface temperature to radiative forcing or something else.

The box plots shown in Fig 6a-c confirm the underestimate of near-surface-temperature Inter-Quartile Range (IQR) at Tiksi (except CAPS), Sodankylä, and Utqiaġvik, and further show that the cold tail of the distribution is generally shorter in the models meaning there is a warm bias during cold periods. The warm bias in cold conditions is well known at Sodankylä and is typical of NWP systems (see Atlaskin and Vihma, 2012 and Day et al., 2020), but this feature has not been shown before at the other two sites to our knowledge.






The models typically also show differences in the distribution of the downwelling radiation at the surface, $LW\downarrow + SW\downarrow$ compared to observations (Fig 6d-f). The IQR is underestimated at Tiksi (except for CAPS) and Utqiaġvik. However, at Sodankylä all the models overestimate the IQR but also do not capture the highest values of incident radiation observed at the top of the distribution. In this study we will not investigate the causes of these discrepancies between the observed and forecast

radiation distributions further, leaving this for a more focussed future study, and will instead move on to focus on the response of the near-surface air temperature and the surface energy budget.










**Fig 6. Boxplots of T2m (a-c) and LW↓+SW↓ (d-f) for Sodankylä, Utqiaġvik and Tiksi in observations and during the second day of the forecast. The text above the boxplots states the median (and inter-quartile-range) of each distribution, which are also shown by the orange line and box edges respectively. The 5-95% range is plotted by the whiskers and points outside this are shown in dots.**

As $LW\!\downarrow +SW_{net}$ is a major driver of 2 m air temperature, errors in 2 m air temperature are either due to errors in this driving term itself, the relationship between $LW\!\downarrow +SW_{net}$ and 2 m temperature, or a more likely combination of both (assuming that errors in advection are negligible). So looking at how the 2m temperature varies as a function of $LW\!\downarrow +SW_{net}$ can provide additional information on the causes of error.



At Sodankylä, Tiksi and Utqiagivk all the models have a conditional warm 2m temperature bias at low levels of incoming radiation ($LW\downarrow +SW_{net}$) (see Fig 7). At Tiksi, Utqiaġvik and Sodankylä the overall sensitivity of T2m to radiative forcing, as measured by the slope of the regression coefficient between 2m-temperature and $LW\downarrow +SW_{net}$ is underestimated in all the models with one exception. The AROME-Arctic model is actually too sensitive at Sodankylä, but captures the observed temperature range at low levels of $LW\downarrow +SW_{net}$.


Note that the LW components used for Sodankylä in this study, are not those provided in the SOP1 MODF, which are collected at the top of the 45m tower, rather they are from a dedicated radiation tower located near the sounding station where the downwelling component is at a height of 16m and the outgoing is at 2m. These were swapped due to a concern over the accuracy of the LW radiation data collected at the met tower (Roberta Pirazzini, personal communication).





**Fig 7: Scatter plots of 2m temperature as a function of $LW{\downarrow} +SW_{net}$ for Sodankylä, Utqiaġvik and Tiksi (from left to right). The regression slope between the 2m temperature and the $LW{\downarrow} +SW_{net}$ is stated in the title, for the observations (in grey) and each model (various colours) during the second day of the forecast.**






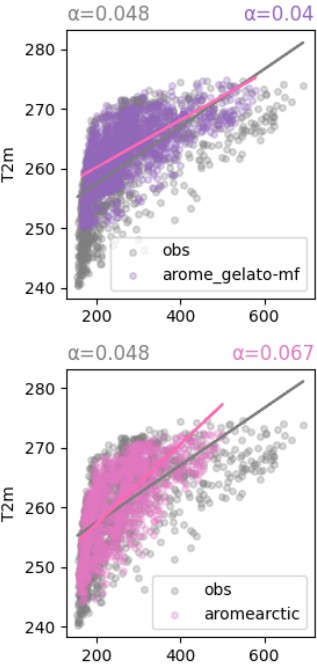


**Fig 7 cont.**

### 3.4 Surface energy budget sensitivity to radiative forcing

Further insight can be gained by constructing surface energy budget sensitivity diagrams, following Miller et al. (2018) and
Day et al. (2020). The idea here is that the surface energy budget can be separated into a "driving term" ($LW\downarrow +SW_{net}$) and
"response terms" (*SHF, LHF, GHF*, and $LW\uparrow$). The relationship between the driving term and each response term can be
summarised with regression coefficients, e.g. for the *SHF*:

$$SHF = \alpha_{SHF}(LW \downarrow +SW_{net}) + \beta_{SHF} \tag{1}$$

where each of the α's can be interpreted as a coupling strength parameter between the driving term and each response term.
These α's provide direct information on the proportional response of each flux term, expressed as a fraction of the total change
in radiative forcing. From this one can see that if, for example, the coupling to the ground heat flux and turbulent fluxes is too
strong in the model (i.e. $|\alpha_{GHF_{mod}} + \alpha_{SHF_{mod}} + \alpha_{LHF_{mod}}| > |\alpha_{GHF_{obs}} + \alpha_{SHF_{obs}} + \alpha_{LHF_{obs}}|$) then $|\alpha_{LW\uparrow}|$ will be too small, i.e.
surface temperature response will be too weak and vice versa. Similarly, compensating errors in the strength of the coupling
to the turbulent fluxes ($\alpha_{SHF_{mod}} + \alpha_{LHF_{mod}}$) and ground heat flux($\alpha_{GHF_{mod}}$) could result in the right surface-temperature
sensitivity, $\alpha_{LW\uparrow}$, but for the wrong reasons. As a result, by comparing the observed and modelled regression coefficients one
can derive physical understanding of the causes of model error.

It is clear from Figures 8, 9 and 10 that all the models generally underestimate the surface temperature sensitivity to radiative
forcing at Sodankylä, Utqiaġvik and Tiksi, because the rate of change in $LW\uparrow$ with changes in radiative forcing, $LW\downarrow +SW_{net}$,
i.e. $\alpha_{LW\uparrow}$ is typically too low (i.e. $\alpha_{LW\uparrow_{mod}} < \alpha_{LW\uparrow_{obs}}$). Since the 2m temperature diagnostic in the models is calculated as a
function of the surface skin temperature, the underestimation of the sensitivity parameter (the regression coefficient) for 2m-
temperature and $LW\uparrow$ and the inability of the models to capture the lowest values of these variables are closely (i.e. comparing
Fig 7 to Figs 8, 9 and 10. For example, at Sodankylä the CAPS model T2m and upwelling longwave ($LW\uparrow$) sensitivities are



very close to what is observed, AROME-Arctic slightly overestimates these sensitivities and SLAV underestimates them. A
similar proportionality can be seen between these properties of the models at the other two sites. Note that because the $LW\uparrow$ at
Sodankylä was observed at 2m and so has rather a small footprint compared to the sensor on the 16m mast, the sensitivity is
more representative of the bare snow than the forest canopy. As a result, one might expect the area mean $LW\uparrow$ sensitivity to
be higher than the value presented here.

This mismatch in terms of $LW\uparrow$ sensitivity goes hand in hand with differences in the other $\alpha$ coefficients and by comparing
the sensitivities of the other response terms in the surface energy budget we can develop some hypotheses about what it leading
to this mismatch in surface temperature sensitivities. For example, at Utqiaġvik, all the models tend to overestimate the
sensitivity of the $GHF$, $\alpha_{GHF}$, which was calculated as the residual of the observed radiative and turbulent fluxes. This can be
an indication of an indication of non-sufficient thermal representation of the land surface, for example lack of a multi-layer
snow model (e.g. Day et al., 2020; Arduini et al., 2019). Unfortunately, we are not able to perform a similar calculation as
performed for Sodankylä, to estimate the $GHF$, as the longwave observations thought to be most reliable, are not co-located
with the other flux observations, or Tiksi, since we don't have the turbulent fluxes in the MODF. As a result, we cannot
calculate the $GHF$ as a residual of the other terms.

Where we have turbulent flux observations, we can also evaluate the $\alpha_{SHF}$ and $\alpha_{LHF}$ terms. At Utqiaġvik, an underestimation
of the sensitivity of the turbulent fluxes, too low $\alpha_{SHF}$ and $\alpha_{LHF}$ in the ARPEGE and SLAV models goes hand in hand with an
overestimation of $\alpha_{GHF}$ mentioned above. In the IFS and ECCC models are closer to observations with smaller values of $\alpha_{GHF}$
and larger values of $\alpha_{SHF}$ and $\alpha_{LHF}$. At Sodankylä, the $\alpha_{SHF}$ varies quite a bit from model to model but all the models where
the LHF was available overestimate the $\alpha_{LHF}$.


At all three sites the relative size of the coefficients varies between the sites, with $\alpha_{LW\uparrow}$, $\alpha_{SHF}$, $\alpha_{GHF}$ typically being an order
of magnitude larger than $\alpha_{LHF}$. This is likely to be typical of cold dry snow-covered environments where the magnitude of the
latent heat flux is low. However, the difference in the relative size of the other three terms varies quite a bit between sites with,
for example, the turbulent flux playing a larger role at Sodankylä than at Tiksi and Utqiaġvik at this time of year. This reflects
the larger surface roughness at Sodankylä associated with the trees at this site.

Before moving on it is worth noting that as well as being used to develop hypotheses about the causes of errors related to the
surface energy budget, these process diagrams and sensitivity metrics could also be applied to test new configurations of NWP
systems with modifications to the land-surface, boundary layer or related schemes and evaluate whether such modifications
are improving the dynamic behaviour with respect to the surface energy budget in line with observed behaviour or not.



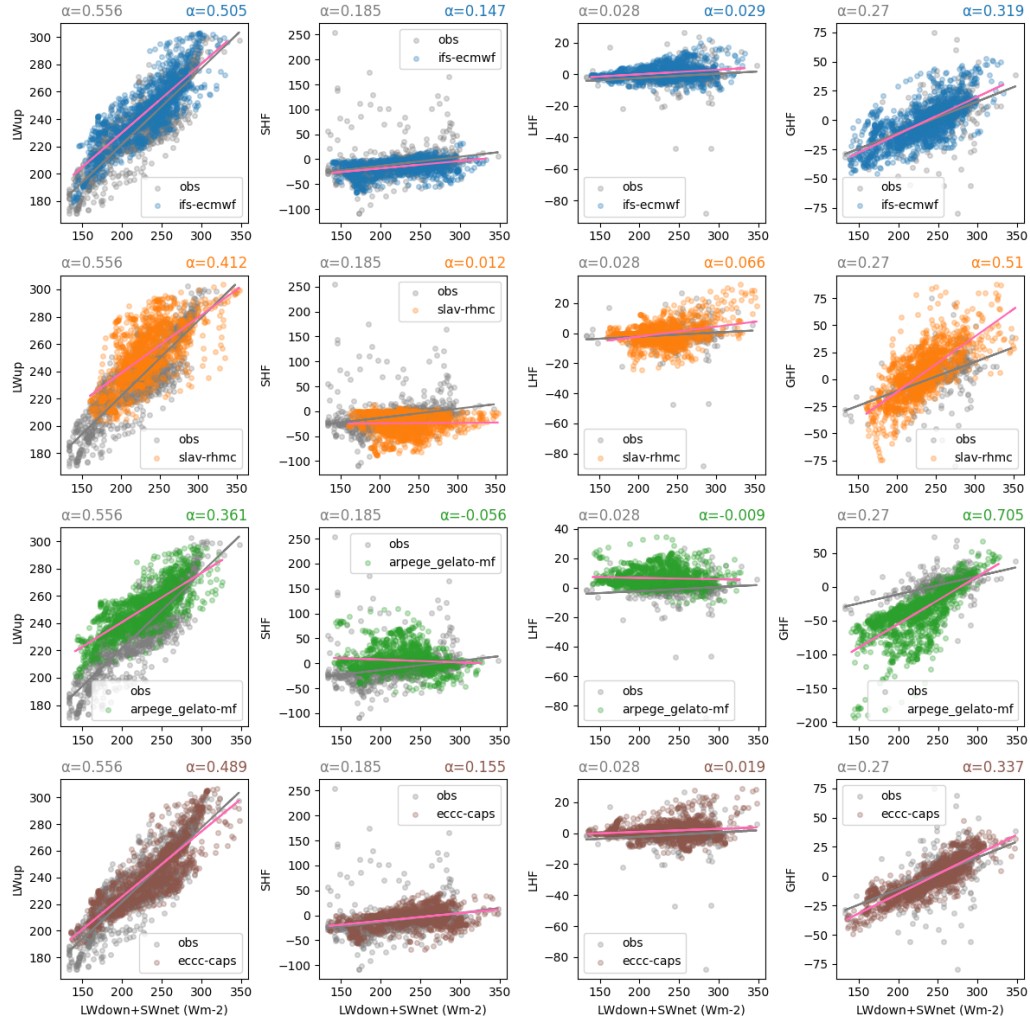


**Figure 8: Process relationship diagrams and sensitivity parameters for upwelling longwave radiation (LWup; left), sensible heat flux (SHF; middle left), latent heat flux (LHF; middle right) and ground heat flux (GHF; right) at Utqiaġvik. Observed values are shown in grey, model values during the second day of the forecast are shown in colour. The line of best linear fit is shown for observations (gray line) and each model (pink line). The sensitivity parameters,**

**a, describing the coupling strength between the driving ($LW\downarrow + SWnet$) and each response term are printed above each diagram, with observational (modelled) relationship on the left (right).**



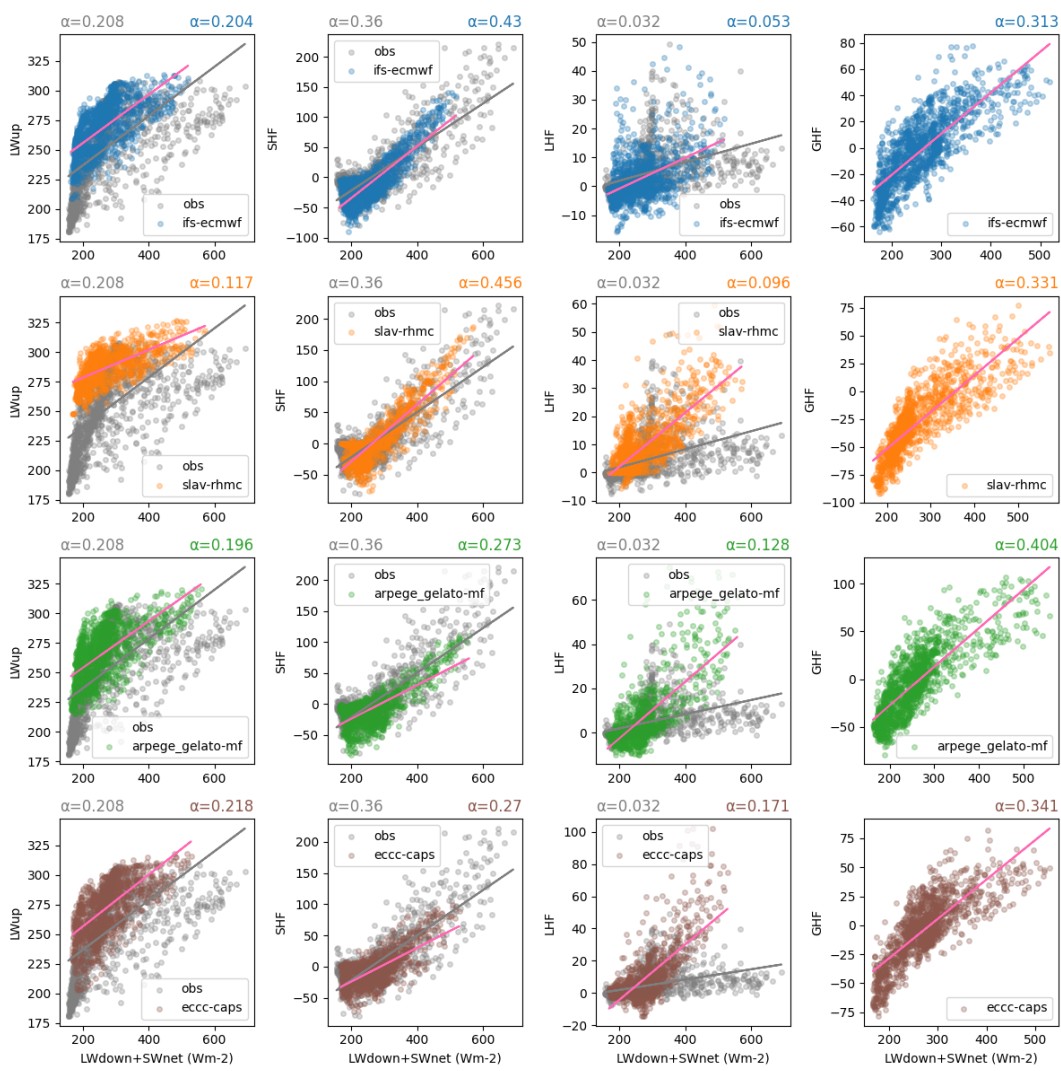

Figure 9: Same as Figure 8 but for Sodankylä.







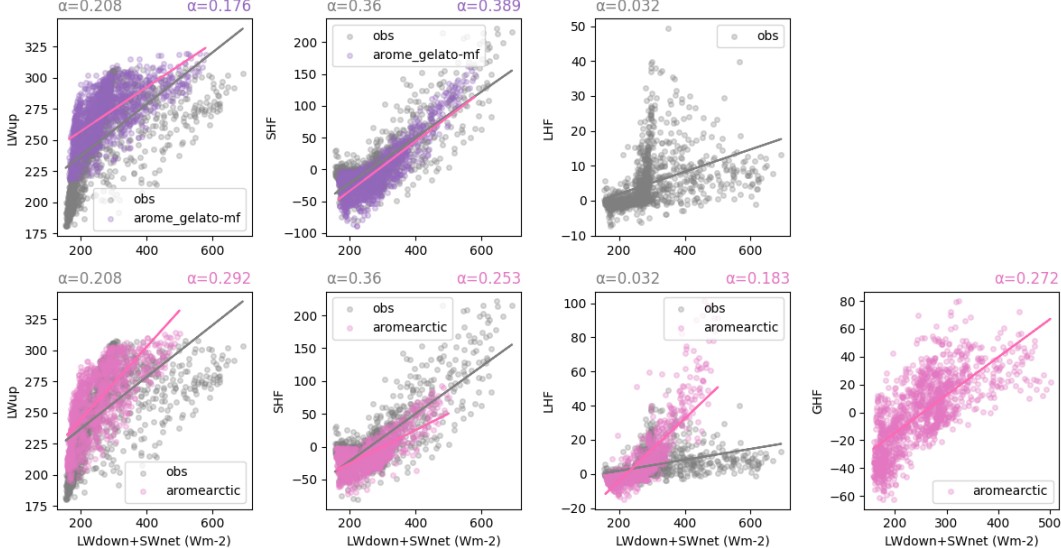

Figure 9: cont.





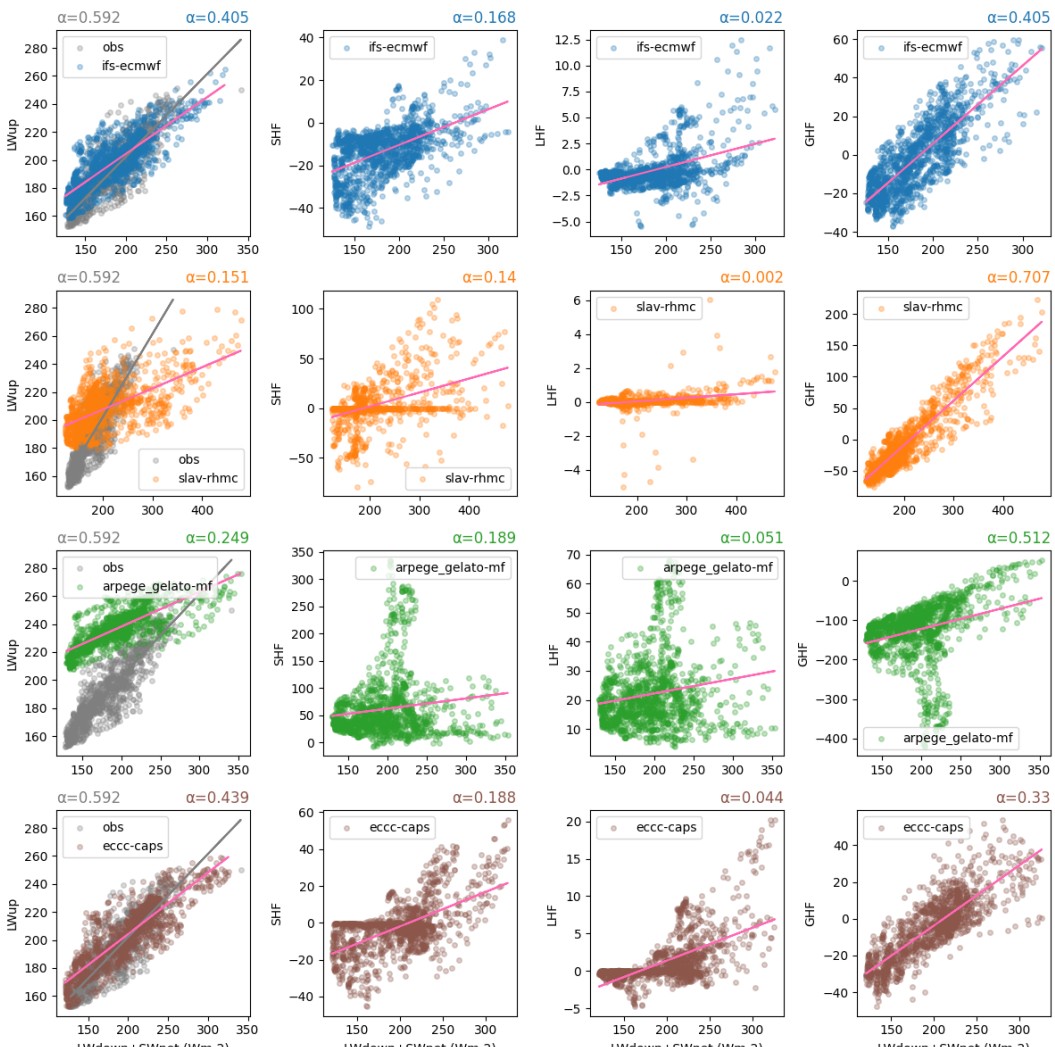

**Figure 10: Same as Figure 8 but for Tiksi.**

### 3.5 Evapuation of wind stress and sensible heat flux

The previous examples highlight discrepancies between forecast and observations and provide hints as to which processes are responsible for the documented errors. The observed conditions also provide multi-variate targets for updated forecasting systems. However, the observations can also help us evaluate a specific process and thereby target a specific parameter or parameterization to change.

The Sodankylä and Utqiaġvik MODFs include turbulent fluxes and profiles of wind speed and temperature allowing us to investigate the parameterisation of turbulent exchanges of heat and momentum at the surface. Turbulent surface fluxes in NWP models are often parameterised according to Monin-Obukhov (M-O) similarity theory where they are related to the gradient in the lowest atmosphere (e.g. Beljaars and Holtslag, 1991):

$$\tau = \rho C_M U_{ref}^2 \tag{2}$$

$$SHF = \rho C_H U_{ref} \left( \theta_{ref} - \theta_{sfc} \right) \tag{3}$$



where $\tau$ is the wind stress, $U$ is the wind speed, $\theta$ is potential temperature, $\rho$ is the air density and the transfer coefficients, $C_M$
and $C_H$, used to in each computation, are a function of the roughness length of momentum and heat, $z_{oM}$ and $z_{oH}$, and a stability
parameter. In these equation the $U_{ref}$ and $\theta_{ref}$ are the wind speed and potential temperature at a reference height, which in the
case of the models is the lowest atmospheric model level, the height of which varies from around 10 to 30m above the surface
depending on the model (see Table 3).

Successfully parameterizing $\tau$ and $SHF$ relies on defining a reasonable function for $C_M$ and $C_H$ and selecting the appropriate
parameters and a proper aggregation of the fluxes in the cases of a tiled surface. Because we have observed and forecast values
for both the fluxes and the bulk parameters in equations 2 and 3 we can diagnose how appropriate the choices in each model
are for the conditions at a particular site. This is done by examining the relationship between the bulk parameters, $U$ and $\theta$,
and the fluxes $\tau$ and $SHF$ (see Figures 11 to 14), as done previously by Tjernström et al. (2005) and more recently by Day et
al. (2020).

In the case of wind stress, in neutral conditions, the points in Figures 11 and 12 would sit on the straight line following:

$$\tau = \rho \frac{k^2 U^2}{\left[ln\left(\frac{z_{ref}}{z_{0M}}\right)\right]^2},$$

where $z_{ref}$ is the height of the lowest model level, $k$ is the von Karman constant and $z_{0m}$ is the aerodynamic roughness length.
The slope of this line is determined by $z_{0m}$. However, this formula provides an overly simplified view as the atmospheric
stability varies from neutral conditions and as a result there is scatter in the values of $\tau$ for any given wind speed.

The relationship between $\tau$ and $U$ for Sodankylä (Figure 11) differs between the models and between the models and the
observations. An estimate of the observed roughness length was also calculated, following the equation above, after selecting
for neutral conditions, and the value is presented in Table 4 along with the value used in each of the models. In the AROME-
Arctic and ICON models, $\tau$ increases too slowly with increasing $U$. This is consistent with the fact that the roughness length
for momentum is too low in these models, which have roughness lengths an order of magnitude lower than that derived from
observations (see Table 4). Increasing $z_{0m}$ in the AROME-Arctic and ICON models would likely reduce the positive bias in
the wind median wind speed profile seen in Figure 4. Interestingly, all models fail to adequately capture the spread of $\tau$ for a
given value of $U$, likely because the models underestimate the atmospheric stability as is suggested by the weaker than observed
thermal stratification indicated by in Figs 4d and 5d. A more detailed study including numerical experimentation would be
needed to demonstrate this further.

At Utqiaġvik, the aerodynamic roughness length is three orders of magnitude lower than at Sodankylä, reflecting the difference
in surface type: snow covered tundra compared to the forested taiga of northern Finland (Table 4). Here the IFS and SLAV
models have roughness lengths close to those derived from observations, whereas the ARPEGE and ICON have values that
are higher. As a result, for a given wind speed the surface stress is too high in these two models (Figure 12).






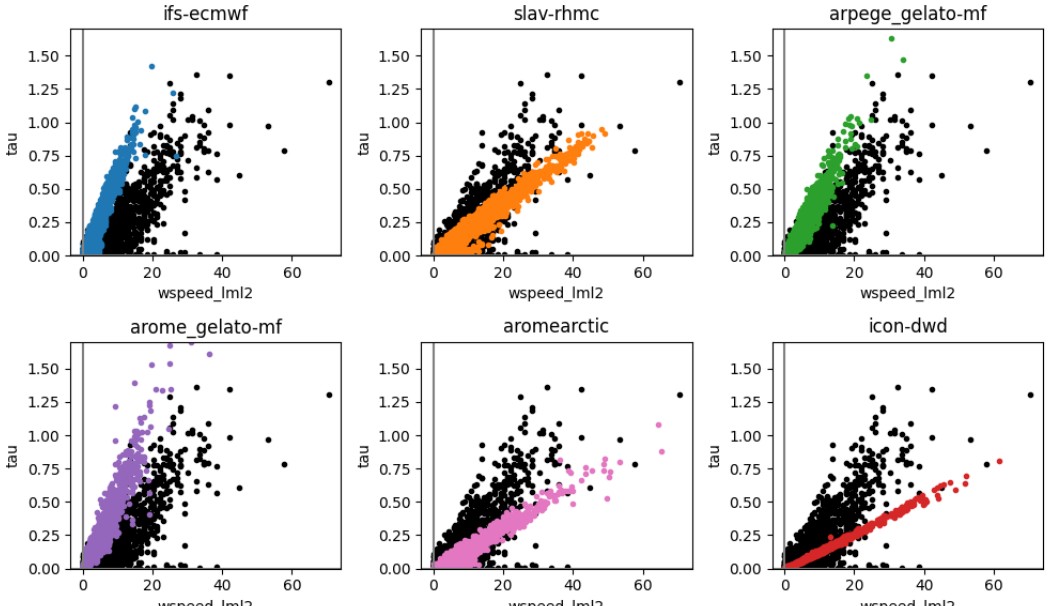

**Figure 11: scatter plots of wind stress vs. the square of the near-surface (lowest model level) wind speed at Sodankylä. The observed points are shown in black and hourly values during the second day of the forecast forecast is shown in colours.**






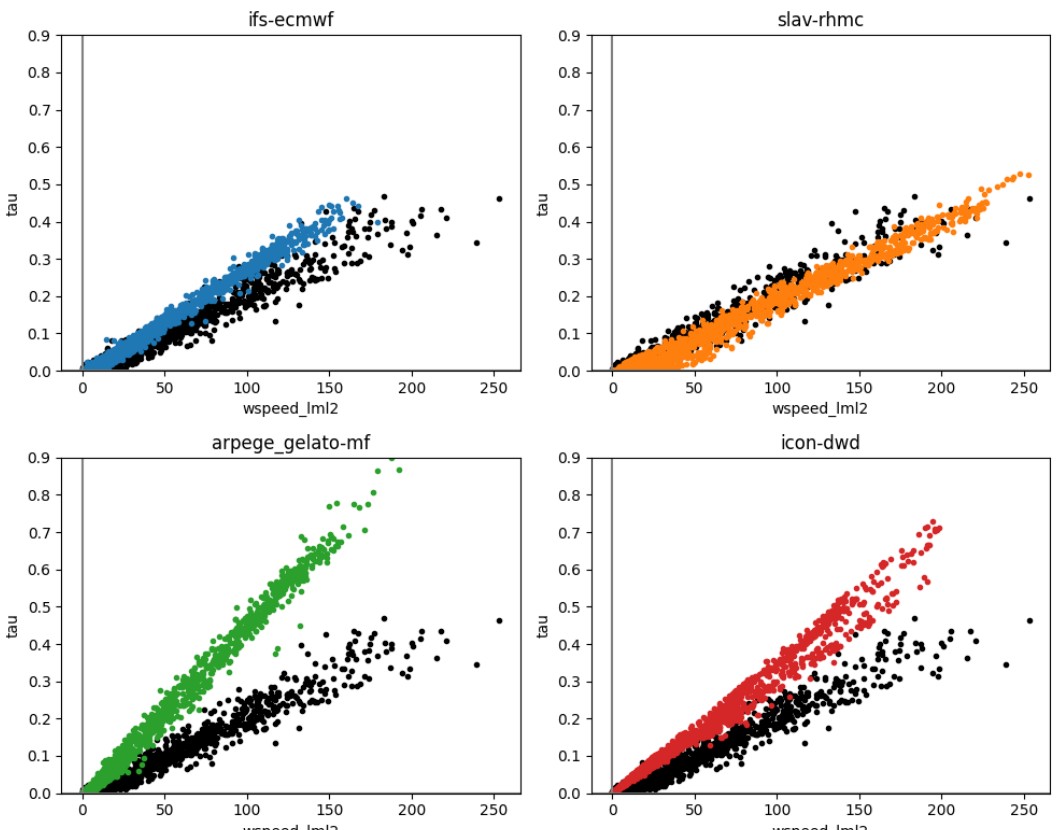

**Figure 12: as Figure 11 but for Utqiaġvik.**

|  | Sodankylä | Utqiaġvik |
|---|---|---|
| Obs | 1.62 | 0.0012 |
| IFS | 1.82 | 0.0013 |
| ARPEGE | 1.50 | 0.0088 |
| SLAV | 1.53 | 0.0013 |
| ICON-DWD | 0.2 | 0.0070 |
| AROME-Arctic | 0.45 | Outside model domain |

**Table 4. Roughness lengths for momentum (m) at Sodankylä and Utqiaġvik from observations and models.**

The scatterplots for the sensible heat flux (Figures 13,14) also provide some insights into the differences in the process representation between the models. The basic shape of the observed points is the same at both sites, but with fewer cases of instability at Utqiaġvik compared to Sodankylä. All the models capture the link between the SHF and the temperature gradient dictated by M-O theory (see Eqn 3) however, the shape of the relationship varies between the models. For example, for the

ARPEGE and AROME-GELATO models the sign of the sensible heat flux does not change in a binary way with $\Delta T$, there is spread in the location along the x-axis where this occurs. This could be due to differences in the numerical formulation of the models, i.e. the timestep at which the flux and temperature terms are stored or due to the fact that we are looking at the gridbox



mean values where the fluxes are aggregated from values computed on different surface tiles. At Sodankylä, the IFS, SLAV and AROME-ARCTIC model have a clear tapering in the sensible heat flux towards zero for high values of $\Delta T$. However, the

AROME-MF, ARPEGE and ICON do not have such a tapering and the scaled heat flux continues to grow with larger $\Delta T$, which is qualitatively inconsistent with the observations and will lead to higher fluxes in very stable conditions inhibiting cooling of the surface. There is also a clear difference in the range of $\Delta T$ between the different models however, in the models this is an aggregate of different surface types representing forest canopy top, bare snow and frozen water and because we do not have a trustable observation of the temperature of the top of the canopy frozen water during freezing conditions it is not

clear what the realistic range should be.

Except for ICON, which has a large fraction of open ocean in the grid cell and therefore are biased towards convective conditions, differences between the models at Utqiaġvik are less pronounced. IFS, SLAV and ARPEGE have quite a similar shape, and all underestimate the magnitude of the scaled heat flux for low values of $\Delta T$, potentially due to the slow bias in

wind speeds near to the surface. Note that the large values of $\Delta T$ for the SLAV model are because the lowest model level is at ~30m, compared to ~10m for the other models.

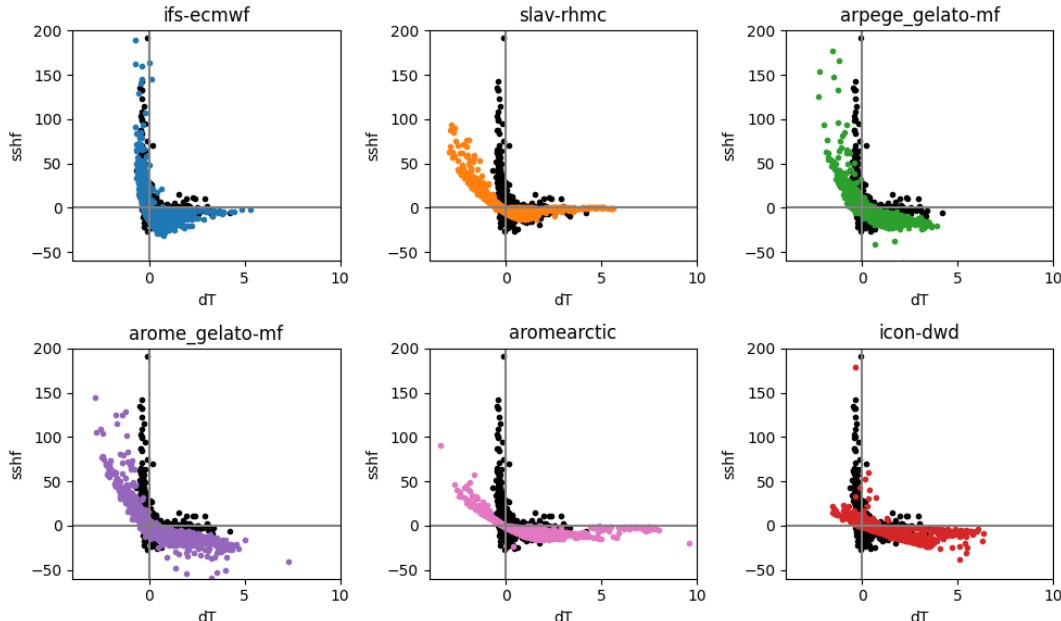

**Figure 13: scatter plots of the scaled sensible heat flux (*SHF/U*) vs. thermal stratification, *ΔT*, at Sodankylä. The observed points are shown in black and hourly values during the second day of the forecasts are shown in colours. Note that the SHF is measured at 24.5 m and for process consistency *ΔT* is calculated using the temperatures observed at 18m and 32m so is not directly comparable with the models.**





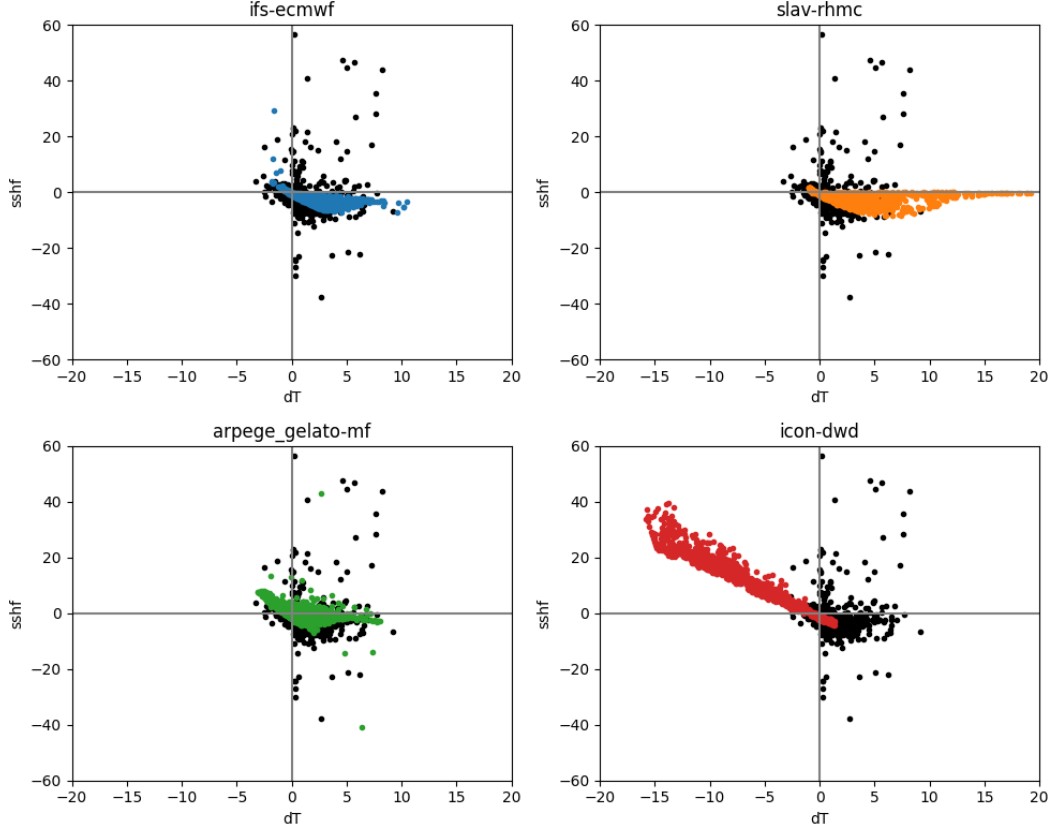


**Figure 14: as Figure 13 but for Utqiaġvik.**

### 4. Conclusions and future plans

In this manuscript we have outlined the motivation for YOPPsiteMIP, documented the current status of the YOPPsiteMIP
forecast data archive on the YOPP data portal (hosted by MET Norway), and presented some multi-model forecast evaluation
examples to demonstrate the utility of the MMDFs and MODFs using data from the YOPP SOP1, which occurred during
February and March 2018. The main conclusions from this analysis are that:

- Near-surface temperature and wind speed forecast errors vary considerably between the different sites, reflecting both
  a range of climate conditions and forecast performance across the geographies represented by this selection of sites.
- A common feature of several sites, namely Sodankylä, Barrow, Tiksi, Eureka, is a conditional warm bias during
    periods of extreme cold which goes hand-in hand with a lack of temperature variability in the lowest ~100m of the
    atmosphere.
  - This lack of variability is investigated further at Utqiaġvik, Tiksi and Sodankylä where radiation components were
    observed and provided which enabled us to investigate the sensitivity of T2m to radiative forcing:
- At all three sites the models tend to underestimate the sensitivity of T2m and the surface skin temperature
      (or $LW{\uparrow}$) to variations in radiative forcing and do not capture extreme minima in these variables, although
      the AROME-Arctic and CAPS models perform better in this regard.



- At Utqiaġvik and Sodankylä, since turbulent fluxes were provided in addition, we were able to investigate the link between these fluxes and the bulk parameters. This highlighted:

   - the thermal representation of the land surface as an issue with all forecasts, likely due to the single-layer representation of snow used in all the forecasts submitted to YOPPsiteMIP but potentially also due to the thermal representation of forest canopy at Sodankylä.
   - Differences in the parameterisation of turbulent fluxes, particularly the specification of the roughness length for momentum which varies by a little less than an order of magnitude between different models.


The development of the MODFs and MMDFs is ongoing and will be completed in phases. The initial phase was to collect basic meteorology data and the main components of the radiation budget. Work on this initial phase is completed and the next phase will provide a wider range of parameters (e.g. turbulent fluxes and cloud parameters) included in the MODFs. This is a more complicated, but very necessary step since the models differ hugely in terms of surface heat and momentum fluxes as

well as cloud properties (not shown). There are also plans to extend the MODF and MMDF concept to Antarctica, focussing on the Southern-hemisphere SOPs. These future phases of the YOPPsiteMIP will allow more detailed studies on e.g.:

- stable boundary layers, diurnal cycles and surface exchange processes,
- cloud radiative forcing and albedo,
- vertical structure of the lower atmosphere,
- assessment of cloud microphysics and hydrometeors,
- assessment of forecast models in Antarctica,
- testing of specific model developments,
- observatory representativeness.

This will allow a more process-focussed understanding of the forecasts in the YOPPsiteMIP archive, but also provide a testbed
for model developers to use when testing new model formulations relevant for the Arctic. Further details on the MODF concept and the SOP1 and 2 MODFs can be found in Uttal et al., (2023) and Morris et al., (2023) respectively. A Python based toolkit for producing the MODFs is available on gitlab and details can be found in Gallagher et al. (manuscript in preparation).

**Appendix A: Table of acronyms**

EDMF=Eddy Diffusivity Mass Flux.
FE=Finite Element,
FD=Finite Difference,
FV=Finite Volume,
H=Hydrostatic,
HARATU = HARMONIE-AROME with RACMO Turbulence
HTESSEL=Hydrology-Tiled ECMWF Scheme for Surface Exchanges over Land,
ICE3 = Three-class ice parameterization
IQR = Inter-Quartile Range
ISBA= Interactions between Surface–Biosphere–Atmosphere,
NH=Non-hydrostatic,
SURFEX = Surface Externalisée,
TERRA = Land Surface module of the ICON weather forecast model.
TKE=Turbulent Kinetic Energy,



**Data availability statement**

All MMDF and MODFs are available on the YOPP Data Portal (https://yopp.met.no), hosted by the Norwegian Meteorological
Institute, for perpetuity (ie. longer than 10 years). The YOPP Data Portal is relying on the Arctic Data Centre
(https://adc.met.no) for data stewarding and the YOPPSiteMIP data can be programmatically accessed using the machine
interface        for        the        Arctic        Data        Centre        or        can        be        accessed        directly        from
https://thredds.met.no/thredds/catalog/alertness/YOPP_supersite/obs/catalog.html,        for        the        MODFs        and
https://thredds.met.no/thredds/catalog/YOPPSiteMIP-models/catalog.html, for the MMDFs.


The SOP1 and SOP2 MODFs for each station shown in white in Fig 1 has been assigned a separate DOI, as described in Morris
et al. (submitted). In the case of the MMDFs a DOI is assigned to the data for each forecast model:

- ECMWF-IFS: https://doi.org/10.21343/A6KA-7142,
- ARPEGE-MF: https://doi.org/10.21343/T31Z-J391,
- SLAV-RHMC: https://doi.org/10.21343/J4SJ-4N61
- DWD-ICON: https://doi.org/10.21343/09KM-BJ07,
- ECCC-CAPS: https://doi.org/10.21343/2BX6-6027,
- AROME-MF: https://doi.org/10.21343/JZH3-2470,
- AROME-Arctic: https://doi.org/10.21343/47AX-MY36.


**Funding**

- JD was supported by European Union's Horizon 2020 Research and Innovation program through Grant Agreement
  871120 (INTERACTIII).

- MT was partially supported with Russian Science Foundation, Grant 21-17-00254
- RP was supported by European Union's Horizon 2020 Research and Innovation program through Grant Agreement
  101003590 (PolarRES)
- TR was supported by the Norwegian Research Council project no. 280573 'Advanced models and weather prediction
  in the Arctic: enhanced capacity from observations and polar process representations (ALERTNESS)'

**Author contributions**

The initial YOPPsiteMIP, MODF and MMDF concepts were developed by GS, JD, BC, TU, SK, LMH, AS and EB. JD, BC,
EB, NA, HF, TR, RF & MT produced or ran simulations to make MMDFs. TU, EA, MG, LXH, JH, ZM, SM, EO, IS, MG, JT
and RP produced or were involved in the production of MODFs. LF, MD and ØG were responsible for the YOPPsiteMIP
archive hosted at MET Norway. JD produced the figures and wrote the manuscript with comments and input from all co-
authors.

**Competing Interests**

The authors declare that they have no conflict of interest.

**Acknowledgements**

This is a contribution to the Year of Polar Prediction (YOPP), a flagship activity of the Polar Prediction Project (PPP), initiated
by the World Weather Research Programme (WWRP) of the World Meteorological Organisation (WMO). We acknowledge
the WMO WWRP for its role in coordinating this international research activity. We would specifically like to thank Thomas
Jung, Jeff Wilson and wider PPP steering group for their tireless support of YOPPsiteMIP.



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
