# Peer review of "The YOPP site Model Intercomparison Project (YOPPsiteMIP) phase 1: project overview and Arctic winter forecast evaluation"

_EGUsphere, 2023_

## Referee Comment (RC1)

Review of
'The YOPP site Model Intercomparison Project (YOPPsiteMIP) phase 1: project overview and
Arctic winter forecast evaluation'
by Day et al.

**General comment :**

This paper presents a numerical weather prediction (NWP) models intercomparison exercise that takes place in the framework of the Year Of Polar Prediction (YOPP) project. This exercice leverages a rich observational dataset of measurements collected at multiple Arctic and Antarctic sites during YOPP special observing periods to evaluate the ability of NWP models to predict temperature, wind and humidity in extreme polar conditions. The intercomparison of many models at many sites has been made possible through the design of a specific file format, the so-called Merged Data Files (MDFs) and Merged Model Data Files (MMDFs), and an associated processing chain in python. The sections 1 and 2 of the paper present all the sites and all the models involved while section 3 presents a first evaluation of models focusing on 7 Arctic sites where data are already available.

I am very impressed by how an international NWP community has collectively – and successfully - designed this ambitious, coordinated intercomparison exercise for polar regions. Collecting all the observations during the YOPP SOPs as well as model forecast from several NWP centers in a common format is already a fantastic achievement.
I am however less convinced by the content of the evaluation itself and by the conclusions drawn about model performances (especially sections 3.3 and 3.4). I think some further work is needed to really show how such a rich intercomparison and evaluation inform about possible shortcomings in the models' physics and dynamics.
In summary, I really would like to see this paper published in GMD but I would also really appreciate the authors to strengthen some parts of the study before, following suggestions herebelow.

**Major comments :**

- Even though the authors mention that details on the sites are provided in Morris et al. (in prep) I think the reader does need some information about the landscape (distance from the coast, relief …) and terrain nature (vegetation cover, snow cover ..) at the different sites. I personally needed such kind of information at many places in the paper at many places in the paper (e.g., l255, l288, l440, l505-515, l569 …). A discussion on the representativity (or non-representativity) of station measurements with respect to the size of model meshes is also needed to disentangle actual model biases from model-observation differences inherent to possible very local nature of the measurements. I admit that adding such information implies increasing the length of the manuscript but a short description of the sites in this paper completed with a critical discussion on the spatial representativity is absolutely necessary to properly follow the analysis and understand the conclusions regarding models' biases.

- The analysis of Figs. 6E-f (line 340-345) is not very conclusive. The authors leave the interpretation of the downward radiative flux biases for a future study but this aspect is essential to correctly understand the reasons behind the surface temperature biases. I would expect at least some additional analysis on the evaluation of the separate distributions of LWdn and SWdn and ideally some conditional analysis between cloudy and non-cloudy scenes. The idea behind this suggestion is to investigate whether models simulate the correct frequency of clouds and if the optical properties thereof is well reproduced.

- L370: I agree that LWdn + SWnet is the effective radiative forcing for the skin surface temperature (and indirectly to 2m temperature, this should be mentioned). Prior to investigate the response of the surface temperature, one first need to know if the albedo at the stations compares well with that observed at the sites (when available).

- L375: Is this due to the inability of models to simulate surface-atmosphere decoupling in clear-sky and windless conditions at those stations? Have you looked at the vertical profiles (simulations vs radiosonde) during these cases?

- L394-397: I do not fully agree here. In convective cases - the main driver of turbulent heat fluxes is indeed the convective instability at the surface driven by radiative forcing. However, in stratified (nocturnal) conditions the main driver of turbulence in the boundary layer (and of the sensible and latent heat fluxes) is the mechanical forcing i.e. the large scale wind speed (Van Hooijdonk et al. 2015, Van de Wiel et al. 2017, Vignon et al. 2017). All the subsequent sensitivity analysis in Sect. 3.4 is therefore incomplete and somewhat misleading for stable conditions. I would strongly recommend the author to carry out the study by separating convective cases from stable cases and to condition the analysis in stable conditions to certain large-scale wind speed classes (or to analyse the dependency of variables upon the large scale wind speed for different classes of LWdn+SWnet).

- Figure 13: In stable conditions, it has been shown that the turbulent heat flux increases then decreases with increasing stability, the maximum value separating a weakly stable from a very stable regime. This behavior is particularly well visible when conditioning the data to conditions with similar radiative forcing (Van Hooijdonk et al 2015). I would have been interested to see if the SHF data at Sodankyla show a clear maximum in stable conditions as well as comments on the ability of models to represent those stable boundary layer regimes (weakly stable cases in cloudy and/or windy conditions versus very stable regime in clear-sky windless conditions).

**Minor comments :**

- Table2: please specify that the timestep is the timestep of the physics (I guess).

- L255: Please recall the model-observation comparison period here.

- Figure 2 and 3: please indicate the local time at the beginning of the x-axes of the station to better identify daytime and nighttime in the graphs. A semi-transparent colour (gray?) shading in the figures themselved during the night periods may also help.

- Figure 5: Are statistics (interquartile ranges) calculated from model data at the same frequency as that of radiosounding?

- L471: Typo 'Evaluation'

- Table 4: Roughness length can vary substantially depending on flow direction, snow cover … please specify the variability ranges as well.

- L535: What is $\Delta T$?

- L546: I realize here that one has to know more specifically for each station which grid point(s) (with which ocean/land percentage) is considered for the evaluation. The information given at lines 141-142 is not sufficient to understand properly this paragraph.

- L557 'T is calculated using the temperatures observed at 18m and 32m so is not directly comparable with the models' This sentence should be included in the main text I think.

- L580: 'likely due to the single-layer representation of snow': This is not shown in the paper, please remove the sentence or rephrase.

- L662: Please remove references to papers in preparation.

van Hooijdonk IGS, Donda JMM, Clercx JH, Bosveld FC, van de Wiel BJH (2015) Shear capacity as prognostic for nocturnal boundary layer regimes. J Atmos Sci 72:1518–1532

van de Wiel BJH, Vignon E, Baas P, van Hooijkdonk IGS, van der Linden SJA, van Hooft JA, Bosveld FC, de Roode SR, Moene AF, Genthon C (2017) Regime transitions in near-surface temperature inversions: a conceptual model. J Atmos Sci 74:1057–1073

Vignon E, van de Wiel BJH, van Hooijdonk IGS, Genthon C, van der Linden SJA, van Hooft JA, Baas P, Maurel W, Traullé O, Casasanta G (2017) Stable boundary layer regimes at dome C, Antarctica: observation and analysis. Q J R Meteorol Soc 143:1241–1253

---

## Author Comment (AC1)

**Response to Reviewers 1 and 2**

**Response to Reviewer 1**
We would like to thank Reviewer 1 for their thoughtful comments and suggestions on the manuscript which have helped us improve it. Reviewer comments are reproduced below in black, with author responses in blue. Line numbers refer to the tracked changes document.

Major comments :
• Even though the authors mention that details on the sites are provided in Morris et al. (in prep) I think the reader does need some information about the landscape (distance from the coast, relief …) and terrain nature (vegetation cover, snow cover ..) at the different sites. I personally needed such kind of information at many places in the paper at many places in the paper (e.g., l255, l288, l440, l505-515, l569 …). A discussion on the representativity (or non-representativity) of station measurements with respect to the size of model meshes is also needed to disentangle actual model biases from model-observation differences inherent to possible very local nature of the measurements. I admit that adding such information implies increasing the length of the manuscript but a short description of the sites in this paper completed with a critical discussion on the spatial representativity is absolutely necessary to properly follow the analysis and understand the conclusions regarding models' biases.

Thanks for raising this important issue, which was also mentioned by Reviewer 2. We have added a new paragraph to the introduction L126-133 to summarise the environment at each site within the manuscript and refer the reader to Mariani (note the change in lead author from Morris) for a more complete description. In order to address the issue of the representativity of the observations of the model grid cell we have selected the model gridpoint that is land-only, if possible. We have also added some discussion of the land-ocean fraction, snow cover and sea ice cover where appropriate (e.g. L159-169).

• The analysis of Figs. 6E-f (line 340-345) is not very conclusive. The authors leave the interpretation of the downward radiative flux biases for a future study but this aspect is essential to correctly understand the reasons behind the surface temperature biases. I would expect at least some additional analysis on the evaluation of the separate distributions of LWdn and SWdn and ideally some conditional analysis between cloudy and non-cloudy scenes. The idea behind this suggestion is to investigate whether models simulate the correct frequency of clouds and if the optical properties thereof is well reproduced.

We agree with the reviewer that the interaction between the clouds and radiation are likely to be very important in understanding the near-surface temperature errors. Unfortunately, for practical reasons it was necessary to omit the cloud information from the first round of MODFs. As a result, it is not possible to perform conditional evaluation of the radiation errors based on the cloud state and we leave this for future analysis. We have added a sentence to this effect at line 424 and mention this as important future work in the conclusions section.

• L370: I agree that LWdn + SWnet is the effective radiative forcing for the skin surface temperature (and indirectly to 2m temperature, this should be mentioned). Prior to investigate the response of the surface temperature, one first need to know if the albedo at the stations compares well with that observed at the sites (when available).

Thanks for this suggestion, we had compared the observed and modelled albedo at the 3 sites but had not included this analysis in the paper. The conclusion was that the albedo looks quite good at all 3 sites, with the modelled values all being within the observed distribution, except the SLAV model has too low an Albedo at Tiksi. This looks to be related to a bias in the fraction of snow, which is only 3% compared to 100% in the IFS for example. This is now included at Figure 7 (Fig 1 in this response) with a short description at L457.

We have also modified the description of the link between the radiative forcing, skin temperature and 2m-temperature as suggested.

[Figure]

**Figure 1. Boxplots of surface albedo for Sodankylä, Utqiaġvik and Tiksi in observations and during the second day of the forecast. The text above the boxplots states the median (and inter-quartile-range) of each distribution, which are also shown by the orange line and box edges respectively. The 5-95% range is plotted by the whiskers and points outside this are shown in dots.**

• L375: Is this due to the inability of models to simulate surface-atmosphere decoupling in clear-sky and windless conditions at those stations? Have you looked at the vertical profiles (simulations vs radiosonde) during these cases?

As mentioned above we cannot stratify the results by cloud cover unfortunately. Looking at the median profiles for the coldest decile one can see that the warm and humid bias extends over the lowest 100m (See Fig 2).

Further if we plot the thermal stratification (Tskin-T$_{\text{lowest model level}}$)/height as a function of the lowest model level wind speed at the 3 sites we see that there is not generally an issue in representing decoupling of Tskin and T10m air-temperature at low windspeeds, We have included this figure and some discussion of it in Section 3.4.

[Figure]

Fig 2. Median profile of lowest 20% of 2m temperature cases at each site. Median temperature (left), specific humidity (middle) and wind speed (right) from the radiosonde (black solid line), the tower (black dashed line), and the numerical models (during the second day of the forecast: colour lines). The mean surface skin temperature is indicated by a dot, 2m temperature (left), 2m specific humidity (middle) and 10m wind speed (right) are shown with a square.

[Figure]

Fig 3. Scatter plots of thermal stratification (T2m-Tlowest model level) as a function of Wind speed on the lowest model at Sodankylä, Utqiaġvik and Tiksi (from left to right) for the observations (in grey) and each model (various colours) during the second day of the forecast.

[Figure]

As Fig 3 but for Sodankyla only.

• L394-397: I do not fully agree here. In convective cases - the main driver of turbulent heat fluxes is indeed the convective instability at the surface driven by radiative forcing. However, in stratified (nocturnal) conditions the main driver of turbulence in the boundary layer (and of the sensible and latent heat fluxes) is the mechanical forcing i.e. the large scale wind speed (Van Hooijdonk et al. 2015, Van de Wiel et al. 2017, Vignon

et al. 2017). All the subsequent sensitivity analysis in Sect. 3.4 is therefore incomplete and somewhat misleading for stable conditions. I would strongly recommend the author to carry out the study by separating convective cases from stable cases and to condition the analysis in stable conditions to certain large-scale wind speed classes (or to analyse the dependency of variables upon the large scale wind speed for different classes of LWdn+SWnet).

We agree with the reviewer that the main driver of boundary layer turbulence in stable conditions is the large-scale wind speed. Indeed, one can see that at Utqiagvik for example, that the turbulent fluxes are almost completely insensitive to the radiative forcing. It is not the intention of this set of sensitivity diagnostics to imply that all of the variability in what we refer to as the "response" variables in the surface energy budget can be explained by the radiative forcing. Rather to condition the fluxes on the radiative forcing specifically and compare with observations to develop further understanding into the role of land-atmosphere exchange processes in the insensitivity of the T2m to radiative forcing. This is why we take a more detailed investigation into the parameterisation of the turbulent fluxes, accounting for the wind speed and thermal stratification in the last results section. We have edited the text in this section to make the motivations and limitations of the diagnostics in this section clearer.

In case of interest, in the supplementary material Day et al. (2020) (https://agupubs.onlinelibrary.wiley.com/doi/10.1029/2020MS002144 ) the fluxes at Sodankyla and Summit, Greenland are plotted as a function of the radiative forcing, but stratified by the Richardson number. One can see, exactly as the reviewer describes, that the sensitivity of the SHF to radiative forcing is low in stable regimes (Ri<0.25), and higher in convective regimes (Ri>0.25).

• Figure 13: In stable conditions, it has been shown that the turbulent heat flux increases then decreases with increasing stability, the maximum value separating a weakly stable from a very stable regime. This behavior is particularly well visible when conditioning the data to conditions with similar radiative forcing (Van Hooijdonk et al 2015). I would have been interested to see if the SHF data at Sodankyla show a clear maximum in stable conditions as well as comments on the ability of models to represent those stable boundary layer regimes (weakly stable cases in cloudy and/or windy conditions versus very stable regime in clear- sky windless conditions).

We have included a figure without the model points obscuring the data and think one can see this maximum between the convective and the very stable cases. So, it does look to be there in the observations. This feature can be seen in the IFS, SLAV-RHMC, Arome-Arctic and ICON models. Arpege and the AROME-MF models do not seem to capture this.

[Figure]

Figure 4: scatter plots of the observed scaled sensible heat flux (SHF/U) vs. thermal stratification, ΔT, at Sodankylä.

Minor comments :
• Table2: please specify that the timestep is the timestep of the physics (I guess).
Done.
• L255: Please recall the model-observation comparison period here.
Done
• Figure 2 and 3: please indicate the local time at the beginning of the x-axes of the station to better identify daytime and nighttime in the graphs. A semi-transparent colour (gray?) shading in the figures themselved during the night periods may also help.
Crosses where the downwelling shortwave is less than 15Wm$^{-2}$ have been added to the plots to indicate night-time as suggested.

• Figure 5: Are statistics (interquartile ranges) calculated from model data at the same frequency as that of radiosounding?
Yes they are. A sentence to communicate this has been added at L353.

• L471: Typo 'Evaluation'
Corrected
• Table 4: Roughness length can vary substantially depending on flow direction, snow cover
… please specify the variability ranges as well.
This is now
• L535: What is ΔT?
ΔT=Tlml-Tskin, where Tlml is the temperature at the lowest model level and Tskin is the surface skin temperature. This is now stated explicitly in the captions of Figs 15 and 16.
• L546: I realize , here that one has to know more specifically for each station which grid point(s) (with which ocean/land percentage) is considered for the evaluation. The information given at lines 141-142 is not sufficient to understand properly this paragraph.
We have compiled the following table with the land fraction at each point and added text from L159 to describe it.

|  | Whitehorse | Iqaluit | Sodankyla | Utqiagvik | Tiksi | Ny-Alesund | Eureka |
|---|---|---|---|---|---|---|---|
| IFS | 100% | 100% | 100% | 100% | 100% | 100% | 100% |
| CAPS | 91.8% | 77.17% | 94.05% | 37.38% | 70.69% | 99.7% | 99.17% |
| ARPEGE | 100% | 100% | 100% | 100% | 100% | 100% | 100% |

| | | | | | | | |
|---|---|---|---|---|---|---|---|
| SLAV | | 100% | 100% | 100% | 100% | 100% | 100% |
| AROMS-MF | | | 100% | | | 100% | |
| ICON | | | 100% | 27% | | 73% | |
| AROME-MN | | | 100% | | | 100% | |

Land fraction within the model gridcell used in the analysis at each site.

• L557 'T is calculated using the temperatures observed at 18m and 32m so is not directly comparable with the models' This sentence should be included in the main text I think.

This has been added to the main text.

• L580: 'likely due to the single-layer representation of snow': This is not shown in the paper, please remove the sentence or rephrase.
• L662: Please remove references to papers in preparation.

References have been updated or removed.

**Response to reviewer 2**

We would like to thank the reviewer for their positive and constructive comments about the manuscript and detailed suggestions for edits, that have helped improve the paper. We provide a point-by-point response below in blue.

General Comments:

This overview of the YOPPsiteMIP project demonstrates the project's value in comparing model output with observations and diagnosing the shortcomings based on the governing equations, heat fluxes for example. The focus is on polar regions that have received less modeling attention than other latitudes. A lot of site-specific information is needed to interpret the results; is this provided in Meta data? This clearly is a major undertaking that has high scientific merit.

We agree with the reviewer, and in response to both reviewers we have provided some additional site-specific information in the introduction at lines 126-133 to aid interpretation of the results but refer to Mariani et al. (2024) for a more complete description of the sites.

I have three questions that should help to further clarify the status of YOPPsiteMIP.

The MODFs are finished for ~50% of Arctic sites and 0% of Antarctic sites (Fig. 1). Why is this?

Thanks for this question. The production of the MODFs was not centralised and different institutions took priority for producing each MODF, i.e. ECCC produced Whitehorse and Iqaluit, NOAA produced Utqiagvik, Eureka and Tiksi, U. Stockholm produced Ny-Alesund and FMI produced Sodankyla. The lack of MODFs for the Antarctic simply reflects a slower uptake of this protocol in that community, but there

are signs this is building some momentum. For example, production of a MODF for the Neymeyer station is under way and will hopefully be published soon.

We have added some additional information at L101 of the introduction to make the status of progress clearer.

How challenging is it to transform the observations and model output into MODFs and MMDFs?

Thanks for this question. It has been quite challenging developing the code to do this and each centre has done their own thing to date, however a python toolkit for producing MODFs is being developed and it is hoped that this toolkit will facilitate a wider uptake of this use of this file format and more timely production of MODFs. This is now mentioned in the conclusions. We hope that this work will be carried forward as part of the WWRP's Polar Coupled Analysis and Prediction for Services Project (PCAPS).

Very high time frequency results are included in MODFs and MMDFs. What is the application of this capability?

Thanks for this question, indeed the motivation for this was omitted from the submitted version of the manuscript. The idea of this was that the averaging can confound effort to understand the model behaviour at the level of individual parameterisations. We have included a sentence on this at line 274. This high frequency data was not utilised in this particular study because a number of the systems only provided hourly output and the focus of this study was on multi-model comparison. It could be used in the future however.

Specific Comments:

Line 55: Need Jung and Matsueda (2014) in reference list.
Done
Lines 74-75: Need details of Gallagher et al. (in prep.) in reference list.
This paper is unlikely to be available before publication so has been removed
Line 173: Correct to "observatories". Also line 241.
Done
Line 181: 2028-06-06?
Changed to 2018
Page 10: Tolstykh et al. (2017) is not in list of references.
Added
Page 11: Under Convection: Kain and Fritsch (1990) is missing from the list of references. Under Microphysics: Correct to Seity et al. (2012).
Done

Page 12: Bastak-Duran et al. (2014) is missing from the list of references.
Done
Line 375: What does conditional mean?
Removed

Line 412: word(s) missing after "closely".

This sentence has be rephrased.

Line 471: "Evaluation".

Changed

Lines 509-510: Positive wind speed bias is seen at Sodankyla for all the models but z0m is reasonable for the other models, so the situation is more complicated than discussed.

In the manuscript we only say that increasing z0m would improve the bias in ICON and Arome-Arctic. We have added some text to explicitly say that this is not the cause of the bias in the other models.

Line 532: "The basic shape of the observed points is the same at both sites". Not true from my perspective.

We have removed this sentence.

Line 546: Correct "are" to "is".

This sentence has been modified.

Lines 664-669: Discriminate between the three Akish and Morris (2023) references.

Done

Lines 706-716: Reposition these references and eliminate duplicates.

Done

Lines 743, 803, 812, 849, 892, 910, 956: References used?

Removed 743, 812, 849, 892, 910 and 956.

Lines 808-811: Discriminate between the two Huang et al. (2023) references.

Done

Move Iacono et al. to correct location.

Moved

Line 908: Move Seifert reference to correct location.

Moved

Lines 945-946: Cox reference is incomplete.

There is no Cox et al., it was the 3rd line of the Uttal et al reference. Edited the line formatting to make this clear.